# ModuLM: Enabling Modular and Multimodal Molecular Relational Learning with Large Language Models

**Zhuo Chen**[1,2], **Yizhen Zheng**[4], **Huan Yee Koh**[4], **Hongxin Xiang**[3], **Linjiang Chen**[5],
**Wenjie Du**[1,2,5,†], **Yang Wang**[1,2,5,†]

[1]University of Science and Technology of China, China
[2]Suzhou Institute for Advanced Research, USTC, China
[3]Hunan University, China    [4]Monash University, Australia
[5]State Key Laboratory of Precision and Intelligent Chemistry, USTC, China
`{czchenzhuo, duwenjie}@mail.ustc.edu.cn`
`{yizhen.zheng1, huan.koh}@monash.edu`
`xianghx@hnu.edu.cn  {linjiangchen, angyan}@ustc.edu.cn`

## Abstract

Molecular Relational Learning (MRL) aims to understand interactions between molecular pairs, playing a critical role in advancing biochemical research. With the recent development of large language models (LLMs), a growing number of studies have explored the integration of MRL with LLMs and achieved promising results. However, the increasing availability of diverse LLMs and molecular structure encoders has significantly expanded the model space, presenting major challenges for benchmarking. Currently, there is no LLM framework that supports both flexible molecular input formats and dynamic architectural switching. To address these challenges, reduce redundant coding, and ensure fair model comparison, we propose ModuLM, a framework designed to support flexible LLM-based model construction and diverse molecular representations. ModuLM provides a rich suite of modular components, including 8 types of 2D molecular graph encoders, 11 types of 3D molecular conformation encoders, 7 types of interaction layers, and 7 mainstream LLM backbones. Owing to its highly flexible model assembly mechanism, ModuLM enables the dynamic construction of over 50,000 distinct model configurations. In addition, we provide comprehensive results to demonstrate the effectiveness of ModuLM in supporting LLM-based MRL tasks. ModuLM is available at `https://github.com/ssjjjhw/ModuLM`.

## 1  Introduction

Molecular Relational Learning (MRL) [36], which aims to understand the interactions between molecular pairs, has garnered growing attention due to its wide-ranging applications across various scientific domains [55]. For example, drug-drug interactions (DDIs) are vital for understanding the effects of concurrent drug use, which can inform strategies to prevent adverse drug reactions and ensure patient safety [42], while solute-solvent interactions (SSIs) are fundamental to solution chemistry and are pivotal in the design and optimization of chemical processes [66, 8, 10]. However, the exhaustive experimental validation of these interactions is notoriously time-consuming and costly.

In recent years, large language models (LLMs) have emerged as a promising new paradigm in MRL research due to their powerful capabilities in knowledge integration and reasoning. Compared with

---

† : corresponding author

traditional methods, LLMs can more efficiently process and understand complex interactions between molecules, significantly improving modeling performance and generalizability. A growing body of research has focused on LLM-based MRL frameworks [52, 29, 17], leveraging the strengths of LLMs to achieve strong results on MRL tasks. For instance, ReactionT5[57] proposed a text-based pretrained LLM tailored for MRL tasks, while MolTC[17] further advanced this line of research by integrating multimodal data and incorporating 2D molecular graphs for improved performance. These developments highlight the research value and application potential of LLMs in MRL. However, with the emergence of an increasing number of encoding methods and backbone models [51, 28, 70, 87, 76, 64, 65, 24], it is now possible to adopt more flexible strategies to recombine components and build more novel model architectures. However, this flexibility introduces new challenges for the benchmarking and evaluation of LLM-based MRL models.

**Lack of diverse input support:** Molecular structures can typically be represented in various forms, such as 1D SMILES strings, 2D molecular graphs, and 3D molecular conformations; however, most existing models support only a single representation modality, commonly 1D SMILES[33, 22, 75] or 2D graphs[20, 59, 9, 12], which limits their ability to fully capture the complexity of molecular interactions and may result in the loss of critical structural information. In the field of LLMs, MRL models that accept 3D molecular structure inputs remain extremely rare, despite the fact that specific 3D conformations are often essential for accurately modeling chemical phenomena, for instance, in small-molecule binding to target proteins[72]. These issues highlight the importance of a unified framework that can accommodate 1D, 2D, and 3D molecular inputs to enable more comprehensive, flexible, and accurate molecular relational learning across a wide range of task scenarios.

**Lack of Flexible Architectures:** Current LLM-based MRL models often adopt relatively rigid architectures. For example, ReactionT5 [57] uses a unified model to encode both SMILES sequences and molecular property descriptions, while MolTC [17] employs graph neural networks (GNNs) to encode molecular graphs. Although these methods achieve performance improvements, they are still somewhat limited by their encoding strategies. In the non-LLM MRL domain, many more effective encoding strategies have been developed, such as MMGNN [11], which employs interpretable GNNs to extract key subgraphs for solvation free energy prediction, and Uni-Mol [84], which introduces pre-trained SE(3) Transformer models specifically designed for molecular data. However, integrating these encoding methods into existing LLM frameworks remains a challenge. Furthermore, most current LLM-based models overlook the modeling of molecular interaction features. These challenges highlight the importance of developing a model framework that can seamlessly combine diverse encoding modules while effectively incorporating molecular interaction information.

To this end, we propose **ModuLM**, a unified and extensible framework designed to overcome the limitations of existing LLM-based MRL approaches. ModuLM provides a highly flexible model construction mechanism that supports a wide range of molecular input formats, multimodal integration strategies, and diverse prompt designs. The framework accepts molecular representations in the form of 1D SMILES, 2D molecular graphs and 3D conformations. It includes 8 types of 2D molecular graph encoders, 11 types of molecular conformation encoders, 7 types of interaction feature encoders, and 7 mainstream LLM backbones, along with specially designed prompt templates for integrating different types of molecular features. To enhance usability and extensibility, ModuLM adopts a modular interface design that allows users to flexibly assemble and extend models, supports incremental pretraining, and is capable of handling complex molecular interaction modeling tasks. ModuLM can generate over 50,000 distinct model configurations. We conduct comprehensive benchmark experiments on tasks such as DDI, SSI and CSI. The results demonstrate that ModuLM performs remarkably well in constructing, evaluating, and comparing LLM-based MRL models. These findings highlight ModuLM's strong potential to advance the development of MRL models and provide valuable insights into molecular interaction mechanisms.

## 2 Related Work

**Molecular Relational Learning:** MRL is critical for drug research, with machine learning offering a scalable alternative to costly experimental validation [37]. Early methods focused on GNNs [33, 82, 78, 18, 71], such as Nyamabo et al.'s substructure-level interaction model using GAT and co-attention [83], and Lee et al.'s CGIB [36], which applies the information bottleneck to extract key substructures. LLM-based methods have gained momentum. For instance, ReactionT5 [53] enhances molecular understanding by integrating chemical structures with natural language. MolTC [17]

further advances this direction by combining 2D molecular graph features with chain-of-thought reasoning to support complex molecular inference.

**LLMs in the Molecular Domain:** LLMs have been widely applied in 1D, 2D, and 3D molecular pattern learning [5]. For 1D, methods like MolT5 [14] and KV-PLM [77] tokenize SMILES strings for representation learning. In 2D, approaches such as Text2Mol [16], MolCA [45], and DrugChat [40] integrate molecular graphs with text encoders or LLMs. For 3D, MolLM [62] and 3D-MoLM [39] incorporate spatial relationships via attention mechanisms and 3D encoders. In addition, LLMs have also found applications in MRL, such as ReactionT5 [57] and MolTC [17], which utilize multimodal data, including molecular graphs (2D), chemical properties, and SMILES (1D), for MRL.

**Deep Learning Frameworks Specialized for MRL:** DeepPurpose is a user-friendly deep learning library for drug-target interaction prediction. It supports customized model training with 15 compound and protein encoders and over 50 neural architectures [27]. FlexMol is another toolkit designed for MRL, offering a variety of encoders and interaction layers that support sequence-based and graph-based representations of drugs and proteins [61].

## 3 ModuLM

In this section, we introduce ModuLM following the model training workflow for LLM-based MRL.

### 3.1 Framework

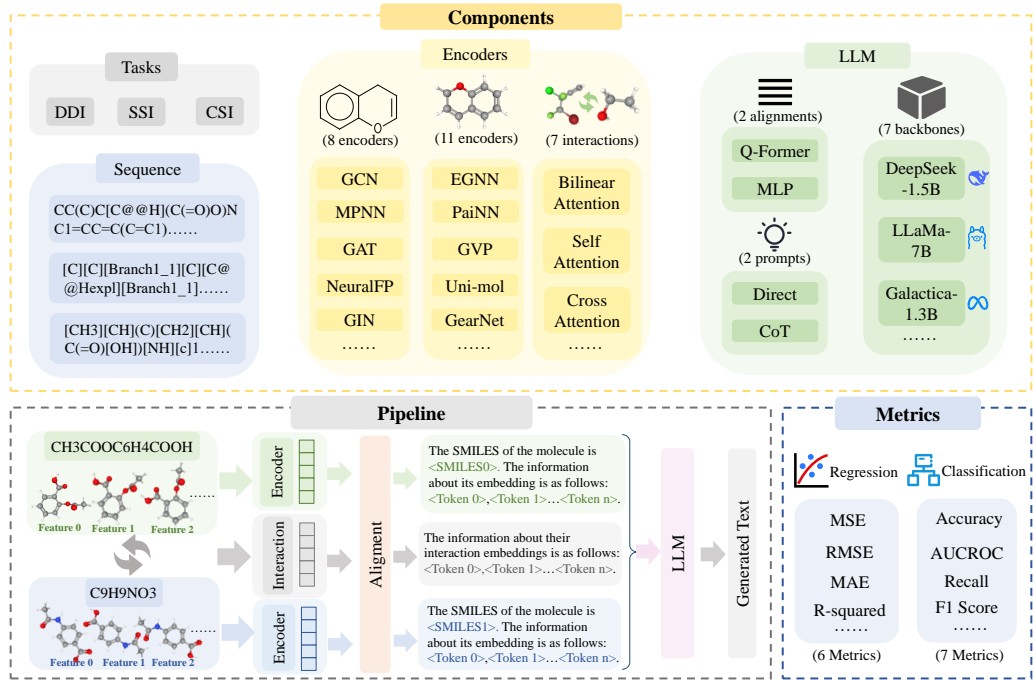

Figure 1: Overview of the ModuLM framework.

ModuLM is an LLM framework designed for MRL, with the overall architecture illustrated in Figure 1. It supports three types of molecular text inputs and accommodates a variety of molecular inputs across 1D, 2D, and 3D modalities. The framework provides eight 2D molecular graph encoders, eleven 3D molecular conformation encoders, and incorporates seven feature interaction designs. It also supports evaluation across multiple task types.

### 3.2 Post Pretraining

To strengthen the domain-specific capabilities of LLMs in chemistry, we begin with incremental pretraining to better adapt them for MRL tasks. In ModuLM, we conduct a survey of various

Table 1: Text settings for different pretraining methods.

---

**Molecular Interaction-based Pretraining**

The first molecule has a SMILES representation of `<SMILES0>`, which suggests certain structural characteristics and chemical functionalities. Based on its structure, it may exhibit the property `[Property0]`, such as high solubility, bioactivity, or specific binding affinity. On the other hand, the second molecule is represented by the SMILES string `<SMILES1>`, and analysis of its structure indicates it may exhibit the property `[Property1]`, potentially contributing to its pharmacokinetic behavior or molecular interaction profile.

---

**Substructure-based Pretraining**

The molecule has a SMILES representation of `<SMILES0>`, which encodes its atomic connectivity and overall molecular structure. It contains notable substructures such as `<substructe0>` and `<substructe1>`, both of which are known to play significant roles in determining the molecule's physicochemical and biological properties. Based on the presence of these functional groups or structural motifs, the molecule is likely to exhibit the properties `[Property0]` and `[Property1]`, which may influence its reactivity, solubility, or interaction with biological targets.

---

authoritative biochemical databases, such as PubChem[1] and DrugBank [32], collecting a large amount of molecular property description texts. We then provide three distinct pretraining strategies: Molecular Interaction-based pretraining, Substructure-based pretraining and Structure Similarity-guided Grouping pretraining.

**Molecular Interaction-based Pretraining** is based on the pretraining method proposed by MolTC [17]. Its pretraining text setup is shown in the Table 1. Considering that molecular interaction tasks typically involve two different molecules, MolTC integrates the textual representations of both molecules and inputs them jointly during pretraining, enabling the LLM to develop prior knowledge in the form of molecular pairs.

**Substructure-based Pretraining** adopts a pretraining text setup as shown in the Table 1, enabling the LLM to learn more fine-grained information within molecules. This allows the LLM to make more accurate judgments based on the potential substructures of different molecules, thereby enhancing its generalization ability when encountering previously unseen molecules and improving performance on downstream tasks.

**Structure Similarity-guided Grouping Pretraining** combines substructure-based pretraining with a grouping strategy based on structural similarity. Molecules with similar structures are grouped together and input as a group during pretraining, thereby enhancing the LLM's understanding of this class of molecules. This method shares the same prompt configuration as the substructure-based pretraining approach, differing in the ordering of the input.

It is worth noting that if you use the Q-former approach for aligning textual and molecular data, additional pretraining will be required. The specific pretraining process can be selected based on your needs. Here, we provide MolTC's [17] methods for aligning molecules with text and molecular data.

### 3.3 Fine-tuning

#### 3.3.1 Input Data

To transform raw molecules into meaningful representations, ModuLM first performs preprocessing followed by encoding. The preprocessing stage includes tasks such as tokenization, normalization, feature extraction, fingerprint generation, molecular graph construction, and molecular conformation generation. The encoding stage is responsible for dynamically constructing the input features for the LLM. In this stage, the preprocessed data is processed to generate embeddings that can be utilized by subsequent network layers. The following will present the specific procedures and methods used by ModuLM to encode different types of data.

**1D Representations of Molecules:** The commonly used molecular representations for MRL in current LLMs are typically SMILES and SelfIES [35]. However, SMILES representations may have

---

[1] https://pubchem.ncbi.nlm.nih.gov

difficulties clearly expressing certain molecular structures. Therefore, for MRL tasks, we recommend using SelfIES. Nonetheless, to provide a more comprehensive benchmarking framework for previous LLM-based approaches, we include both molecular text encoding methods here, and additionally incorporate SMARTS. For 1D representations of molecules, the encoding method depends on the specific LLM used.

**2D Molecular Graph Representation:** Since 1D molecular representations often fail to capture the structural information of molecules, it is common in MRL tasks to incorporate multimodal molecular structure information. In non-LLM research areas, molecular graph representation is the most widely used approach. In most tasks, molecular graphs help models better understand molecular structures, thereby enhancing MRL performance. We provide a variety of methods as shown in the Table 2.

**3D Molecular Conformation Representation:** In most MRL tasks, 2D molecular graphs are sufficient for the model to make effective judgments. However, in certain specialized tasks, it is necessary to incorporate the 3D spatial information of molecules to accurately determine intermolecular interactions. This is an aspect that current LLM-based MRL models have largely overlooked. In ModuLM, we address this limitation by providing various 3D structure encoding methods to support more advanced molecular relation learning. The encoding methods are listed in the Table 2.

**Interaction:** In current LLM-based MRL model designs, the explicit modeling of molecular interaction relationships is often overlooked. In ModuLM, we address this issue by introducing specially designed prompts and interaction layers to incorporate interaction information into the LLM. The Interaction Layer is another key building block of the MRL model. These layers serve two main functions: capturing and modeling the relationships between different molecular entities, and integrating multiple embeddings of the same entity to form a more comprehensive feature representation. The Interaction Layer can accept encoded inputs from different molecules, enabling the construction of more complex model architectures. In ModuLM, the interaction designs we provide are shown in Table 2.

Table 2: Encoding methods for different data formats

| Encoder Type | Methods |
| --- | --- |
| **2D Graph** | GCN [34], MPNN [22], GAT [68], NeuralFP [13], AttentiveFP [73], GIN [74], GraphSAGE[25], CoATGIN[80] |
| **3D Conformation** | EGNN [58], 3D-GeoFormer [86], SE3Transformer [19], PaiNN [59], GVP [30], GearNet [81], DimeNet++ [21], SchNet [60], SphereNet [44], G-SphereNet[46], Uni-mol [85] |
| **Interaction** | Bilinear Attention[2], Self Attention[67], Cross Attention[54], Highway[87], Gated Fusion[47], Bilinear Fusion[41], Mean |

### 3.3.2 Alignment

Since data from different modalities typically exist in distinct semantic spaces, it is necessary to perform alignment before feeding them into the LLM. Currently, two common approaches are used: employing a lightweight MLP [43] or using a Q-former [38]. In the ModuLM framework we provide, both alignment methods are supported. It is worth noting that when using a Q-former, it is typically involved during the pretraining stage to enable better alignment performance.

### 3.3.3 Backbone

Existing LLM-based MRL models often adopt different backbone architectures, and there has been no systematic investigation into the MRL performance across different LLM backbones. In ModuLM, we provide a streamlined method for switching backbones. We have surveyed and integrated a range of mainstream LLMs and offer simple interfaces for replacement, enabling easier comparison of performance differences across various types and scales of LLMs under a unified experimental setup. Here, we provide two types of prompts: direct inference and chain-of-thought-based reasoning. The specific prompt designs are detailed in the Appendix A.3.5.

## 3.4 Evaluation Metrics

ModuLM supports multiple default metrics, aligning with the TDC standard for molecular relational learning[26]. Users can specify the metrics in the Trainer for early stopping and testing. These metrics include various regression metrics (Mean Squared Error (MSE), Root-Mean Squared Error (RMSE), Mean Absolute Error (MAE), Coefficient of Determination ($R^2$), Pearson Correlation Coefficient (PCC), Spearman Correlation Coefficient), binary classification metrics (Area Under Receiver Operating Characteristic Curve (AUC-ROC), Area Under the Precision-Recall Curve (PR-AUC), Range LogAUC, Accuracy Metrics, Precision, Recall, F1 Score.

## 3.5 Supporting Datasets

ModuLM is compatible with all MRL datasets that conform to our specified format. These datasets typically consist of three components: molecular entity one, molecular entity two, and a label. We provide utility functions to facilitate the loading of datasets in this format. In addition, ModuLM includes a wide range of built-in datasets from various domains, such as Drugbank (Version 5.0.3), ZhangDDI [79], ChChMiner [88], DeepDDI [56], TWOSIDES [63], Chromophore [31], MNSol [48], CompSol [50], Abraham [23], CombiSolv [69], FreeSolv [49], and CombiSolv-QM [69]. For more details, please refer to the Appendix A.2.

# 4 Experiments

We conduct validation experiments on MRL tasks using ModuLM to demonstrate the framework's capability in supporting a wide range of experiments, comparisons, and analyses. The following sections present the results for DDI tasks, showcasing the impact of different inputs and encoders on the performance of LLMs in MRL. More experimental details are provided in the Appendix A.5.

## 4.1 Experimental Setup

Table 3: Experimental Settings on DDI Datasets

| Experiment No. | Backbone | Encoder | Interaction | Input Feature |
|---|---|---|---|---|
| 1.1 | Galactica-1.3B | - | - | $m_s$ |
| 1.2 | Galactica-1.3B | GIN | - | $m_s + m_g$ |
| 1.3 | Galactica-1.3B | GIN | Cross Attention | $m_s + m_g$ |
| 1.4 | Galactica-1.3B | Uni-mol | - | $m_s + m_c$ |
| 1.5 | Galactica-6.7B | MPNN | Gated Fusion | $m_s + m_g$ |
| 1.6 | DeepSeek-1.5B | - | - | $m_s$ |
| 1.7 | DeepSeek-1.5B | GIN | - | $m_s + m_g$ |
| 1.8 | DeepSeek-1.5B | Uni-mol | - | $m_s + m_c$ |
| 1.9 | DeepSeek-7B | 3D-GeoFormer | Highway | $m_s + m_c$ |
| 1.10 | DeepSeek-14B | Uni-mol | - | $m_s + m_c$ |
| 1.11 | DeepSeek-14B | GAT | Self Attention | $m_s + m_g$ |
| 1.12 | LLaMA-1B | - | - | $m_s$ |
| 1.13 | LLaMA-1B | CoATGIN | - | $m_s + m_g$ |
| 1.14 | LLaMA-1B | EGNN | Gated Fusion | $m_s + m_c$ |
| 1.15 | LLaMA-13B | SchNet | Bilinear Attention | $m_s + m_c$ |

**Note:** $m_s$ = molecular sequence, $m_g$ = molecular graph, $m_c$ = molecular conformation. '-' indicates that no method is applied.

Given the flexibility of ModuLM, which enables a large number of potential model combinations, the goal of this section is not to exhaustively explore the entire model space. Instead, we select several model combinations as examples to demonstrate ModuLM's robust capabilities in constructing and evaluating diverse model architectures across various datasets and performance metrics. It is worth noting that for different backbones, we adopt a unified pretraining strategy. The experimental backbones presented in the main text are LLMs pretrained based on the Structure Similarity-guided Grouping pretraining approach.

During the evaluation phase on downstream tasks, we utilized the same datasets used in the MolTC framework [17] for evaluating MRL tasks, including DrugBank (Version 5.0.3), ZhangDDI [79], ChChMiner [88], DeepDDI [56], TWOSIDES [63], Chromophore [31], MNSol [48], CompSol [50], Abraham [23], CombiSolv [69], FreeSolv [49], and CombiSolv-QM [69]. We further process these datasets using RDKit by generating multiple conformations for each molecule, aiming to explore the performance of ModuLM on 3D conformation-based LLM-driven MRL tasks. Based on ModuLM, we construct 15 models, as detailed in Table 3, and compare them with five state-of-the-art LLM-based models for MRL tasks: Galactica, ChemT5 [6], MolT5, MolCA [45], and MolTC [17].

For the DDI, SSI and CSI datasets, we randomly split the data into training, validation, and testing sets in a ratio of 7:2:1. Each experiment was repeated five times to mitigate the effects of randomness, and the average results were reported. All experiments were conducted using eight NVIDIA A100 80G GPUs. For more specific training details, please refer to the Appendix A.5.

## 4.2 Experimental Results and Analysis

Table 4: Performance on DDI Datasets

| Experiment | AUC-ROC (ChChMiner) | Accuracy (ChChMiner) | AUC-ROC (ZhangDDI) | Accuracy (ZhangDDI) | AUC-ROC (DeepDDI) | Accuracy (DeepDDI) |
|---|---|---|---|---|---|---|
| Chem T5[7] | $0.867 \pm 0.012$ | $0.814 \pm 0.009$ | $0.889 \pm 0.017$ | $0.751 \pm 0.021$ | $0.856 \pm 0.012$ | $0.784 \pm 0.013$ |
| MolCA[45] | $0.924 \pm 0.006$ | $0.901 \pm 0.009$ | $0.895 \pm 0.006$ | $0.745 \pm 0.010$ | $0.878 \pm 0.014$ | $0.841 \pm 0.015$ |
| MolT5[15] | $0.914 \pm 0.019$ | $0.862 \pm 0.022$ | $0.901 \pm 0.011$ | $0.802 \pm 0.015$ | $0.907 \pm 0.014$ | $0.870 \pm 0.016$ |
| MolTC[17] | $0.964 \pm 0.008$ | $0.957 \pm 0.006$ | $\mathbf{0.941 \pm 0.006}$ | $0.896 \pm 0.008$ | $\mathbf{0.977 \pm 0.013}$ | $0.956 \pm 0.011$ |
| 1.1 | $0.933 \pm 0.011$ | $0.924 \pm 0.009$ | $0.912 \pm 0.008$ | $0.854 \pm 0.004$ | $0.899 \pm 0.010$ | $0.855 \pm 0.009$ |
| 1.2 | $0.956 \pm 0.008$ | $0.943 \pm 0.009$ | $0.930 \pm 0.006$ | $0.872 \pm 0.009$ | $0.924 \pm 0.008$ | $0.887 \pm 0.008$ |
| 1.3 | $0.960 \pm 0.010$ | $0.954 \pm 0.006$ | $0.933 \pm 0.007$ | $0.891 \pm 0.004$ | $0.939 \pm 0.007$ | $0.904 \pm 0.008$ |
| 1.4 | $0.955 \pm 0.005$ | $0.949 \pm 0.008$ | $0.936 \pm 0.014$ | $0.901 \pm 0.010$ | $0.956 \pm 0.008$ | $0.919 \pm 0.007$ |
| 1.5 | $0.940 \pm 0.009$ | $0.932 \pm 0.008$ | $0.921 \pm 0.008$ | $0.866 \pm 0.005$ | $0.948 \pm 0.009$ | $0.912 \pm 0.006$ |
| 1.6 | $0.936 \pm 0.010$ | $0.930 \pm 0.011$ | $0.920 \pm 0.009$ | $0.860 \pm 0.008$ | $0.906 \pm 0.008$ | $0.872 \pm 0.008$ |
| 1.7 | $0.957 \pm 0.008$ | $0.953 \pm 0.010$ | $0.934 \pm 0.006$ | $0.889 \pm 0.004$ | $0.958 \pm 0.007$ | $0.942 \pm 0.007$ |
| 1.8 | $\mathbf{0.966 \pm 0.007}$ | $\mathbf{0.964 \pm 0.005}$ | $0.938 \pm 0.005$ | $\mathbf{0.907 \pm 0.006}$ | $0.972 \pm 0.009$ | $\mathbf{0.959 \pm 0.010}$ |
| 1.9 | $0.944 \pm 0.010$ | $0.935 \pm 0.009$ | $0.925 \pm 0.005$ | $0.870 \pm 0.003$ | $0.955 \pm 0.008$ | $0.930 \pm 0.007$ |
| 1.10 | $0.931 \pm 0.012$ | $0.918 \pm 0.010$ | $0.916 \pm 0.009$ | $0.861 \pm 0.011$ | $0.943 \pm 0.010$ | $0.915 \pm 0.008$ |
| 1.11 | $0.935 \pm 0.007$ | $0.921 \pm 0.012$ | $0.906 \pm 0.009$ | $0.855 \pm 0.008$ | $0.936 \pm 0.009$ | $0.908 \pm 0.007$ |
| 1.12 | $0.925 \pm 0.013$ | $0.911 \pm 0.011$ | $0.901 \pm 0.008$ | $0.852 \pm 0.006$ | $0.895 \pm 0.010$ | $0.850 \pm 0.009$ |
| 1.13 | $0.945 \pm 0.009$ | $0.937 \pm 0.008$ | $0.925 \pm 0.007$ | $0.870 \pm 0.006$ | $0.935 \pm 0.008$ | $0.904 \pm 0.008$ |
| 1.14 | $0.951 \pm 0.007$ | $0.946 \pm 0.011$ | $0.928 \pm 0.004$ | $0.875 \pm 0.005$ | $0.946 \pm 0.007$ | $0.918 \pm 0.006$ |
| 1.15 | $0.915 \pm 0.016$ | $0.896 \pm 0.013$ | $0.913 \pm 0.008$ | $0.860 \pm 0.002$ | $0.928 \pm 0.011$ | $0.897 \pm 0.008$ |

Table 5: Performance on SSI Datasets

| Experiment | MAE (FreeSolv) | RMSE (FreeSolv) | MAE (CompSol) | RMSE (CompSol) | MAE (CombiSolv) | RMSE (CombiSolv) |
|---|---|---|---|---|---|---|
| Chem T5[7] | $0.923 \pm 0.022$ | $1.511 \pm 0.043$ | $0.611 \pm 0.017$ | $0.766 \pm 0.032$ | $0.840 \pm 0.040$ | $1.294 \pm 0.043$ |
| MolCA[45] | $0.761 \pm 0.034$ | $1.303 \pm 0.039$ | $0.505 \pm 0.036$ | $0.726 \pm 0.040$ | $0.771 \pm 0.033$ | $1.130 \pm 0.027$ |
| MolT5[15] | $0.733 \pm 0.047$ | $1.135 \pm 0.059$ | $0.496 \pm 0.028$ | $0.708 \pm 0.020$ | $0.677 \pm 0.024$ | $1.066 \pm 0.027$ |
| MolTC[17] | $0.533 \pm 0.018$ | $0.726 \pm 0.022$ | $0.244 \pm 0.018$ | $0.356 \pm 0.022$ | $0.237 \pm 0.019$ | $0.465 \pm 0.022$ |
| 1.1 | $0.710 \pm 0.021$ | $1.120 \pm 0.030$ | $0.472 \pm 0.024$ | $0.665 \pm 0.028$ | $0.615 \pm 0.026$ | $0.984 \pm 0.032$ |
| 1.2 | $0.570 \pm 0.020$ | $0.910 \pm 0.028$ | $0.384 \pm 0.022$ | $0.540 \pm 0.025$ | $0.568 \pm 0.023$ | $0.930 \pm 0.030$ |
| 1.3 | $0.556 \pm 0.018$ | $0.840 \pm 0.025$ | $0.366 \pm 0.021$ | $0.522 \pm 0.024$ | $0.487 \pm 0.021$ | $0.820 \pm 0.027$ |
| 1.4 | $0.534 \pm 0.017$ | $0.808 \pm 0.024$ | $0.347 \pm 0.020$ | $0.501 \pm 0.023$ | $0.447 \pm 0.020$ | $0.780 \pm 0.026$ |
| 1.5 | $0.580 \pm 0.016$ | $0.972 \pm 0.023$ | $0.403 \pm 0.019$ | $0.575 \pm 0.021$ | $0.579 \pm 0.019$ | $0.898 \pm 0.025$ |
| 1.6 | $0.685 \pm 0.015$ | $1.086 \pm 0.022$ | $0.451 \pm 0.018$ | $0.643 \pm 0.020$ | $0.602 \pm 0.018$ | $0.945 \pm 0.024$ |
| 1.7 | $0.550 \pm 0.014$ | $0.749 \pm 0.021$ | $0.271 \pm 0.017$ | $0.415 \pm 0.019$ | $0.289 \pm 0.017$ | $0.515 \pm 0.023$ |
| 1.8 | $\mathbf{0.510 \pm 0.013}$ | $\mathbf{0.698 \pm 0.020}$ | $\mathbf{0.191 \pm 0.016}$ | $\mathbf{0.298 \pm 0.018}$ | $\mathbf{0.190 \pm 0.016}$ | $\mathbf{0.388 \pm 0.022}$ |
| 1.9 | $0.555 \pm 0.014$ | $0.825 \pm 0.021$ | $0.335 \pm 0.017$ | $0.490 \pm 0.019$ | $0.393 \pm 0.017$ | $0.595 \pm 0.023$ |
| 1.10 | $0.605 \pm 0.015$ | $0.850 \pm 0.022$ | $0.443 \pm 0.018$ | $0.598 \pm 0.020$ | $0.548 \pm 0.018$ | $0.864 \pm 0.024$ |
| 1.11 | $0.590 \pm 0.016$ | $0.876 \pm 0.023$ | $0.458 \pm 0.019$ | $0.601 \pm 0.021$ | $0.556 \pm 0.019$ | $0.839 \pm 0.025$ |
| 1.12 | $0.745 \pm 0.017$ | $1.091 \pm 0.024$ | $0.514 \pm 0.020$ | $0.692 \pm 0.022$ | $0.687 \pm 0.020$ | $1.008 \pm 0.026$ |
| 1.13 | $0.605 \pm 0.016$ | $0.880 \pm 0.023$ | $0.374 \pm 0.019$ | $0.530 \pm 0.021$ | $0.538 \pm 0.019$ | $0.820 \pm 0.025$ |
| 1.14 | $0.580 \pm 0.015$ | $0.887 \pm 0.022$ | $0.360 \pm 0.018$ | $0.515 \pm 0.020$ | $0.460 \pm 0.018$ | $0.790 \pm 0.024$ |
| 1.15 | $0.630 \pm 0.018$ | $0.910 \pm 0.025$ | $0.450 \pm 0.021$ | $0.633 \pm 0.023$ | $0.567 \pm 0.021$ | $0.892 \pm 0.027$ |

Table 4 and Table 5 show the experimental results of commonly used MRL LLMs, as well as their performance under our experimental settings. It is worth noting that in our experimental setup, we adopt a direct inference approach using LLMs without employing chain-of-thought reasoning. In our setup, we design models with different backbones, input formats, and encoders. Due to space limitations, additional experimental results are presented in the Appendix A.5.6.

**Impact of Input Data and Encoders:** We use the settings {1.1, 1.2, 1.4}, {1.6, 1.7, 1.8}, and {1.12, 1.13} to evaluate ModuLM's ability to analyze and integrate different inputs and encoder layers within LLMs. The results in the table indicate that integrating multimodal information, such as 2D molecular graphs and 3D molecular conformations, can indeed enhance model performance. Notably, models that incorporate 3D molecular conformation information achieve the best results.

**Impact of Interaction Layers:** We use the configurations {1.3, 1.5, 1.9, 1.11, 1.14, 1.15} to evaluate ModuLM's ability to analyze the effect of feature interaction layers. We first conduct experiments with various non-interaction designs. The analysis shows that adding interaction layers consistently improves model performance to some extent. This confirms the importance of interaction layers in LLM-based MRL models. However, existing LLM comparisons generally ignore multi-molecular interaction information. ModuLM enables multi-dimensional analysis, which helps better assess the impact of different types of multimodal information on model performance.

**Impact of Different Backbones:** Existing LLM-based MRL tasks generally lack systematic evaluation across different backbones. However, thoroughly testing LLMs under various experimental configurations requires significant time and resources. ModuLM offers a framework for efficient and rapid evaluation. As shown in Table 4 and Table 5, we conduct experiments using different backbones. Among them, models from the DeepSeek series often achieve better performance. Interestingly, our results reveal that larger model sizes do not necessarily lead to better performance in MRL tasks. This may be because larger LLMs possess stronger generalization ability, which can limit task-specific adaptation during fine-tuning. In contrast, smaller LLMs adapt better during fine-tuning, leading to stronger task specialization in MRL scenarios.

## 4.3  Custom Model Design and Evaluation

This section demonstrates how users can leverage ModuLM to extend, construct, and analyze more complex models. Figure 2 presents an overview of the newly proposed model architecture. This

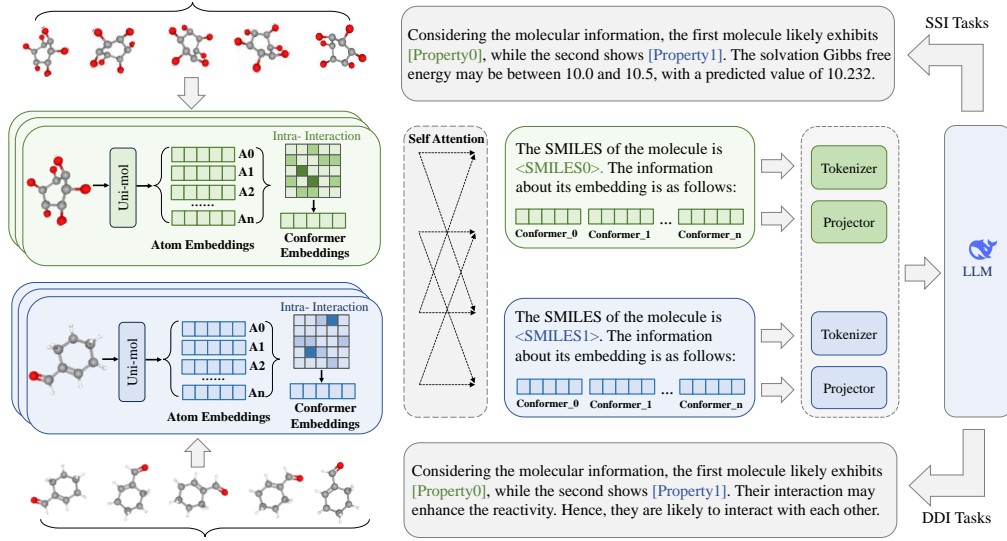

Figure 2: An overview of the custom model designed using ModuLM.

design builds upon the best-performing configuration identified in Experiment 1.8, which utilizes 3D molecular conformations, employs Uni-Mol as the encoder, and adopts DeepSeek-1.5B as the LLM backbone. In this enhanced version, we further refined the encoder and the dual-molecule interaction mechanism, and incorporated CoT prompting strategy to boost the model's reasoning capabilities.

To simulate practical scenarios in which users may wish to combine predefined encoders with custom modules, we reimplement the molecular encoder as a user-defined component and integrate it with other existing encoder modules to construct a complete, functional model. Specifically, we redesign the representation of atom-level data encoded by Uni-Mol by calculating the attention weights of each

atom to all other atoms within the conformation. These weights are then used to reaggregate the atom features, and the mean of the weighted features serves as the new conformation-level representation. The full model construction process based on ModuLM is detailed in Appendix A.4.

We retained the experimental settings described in Section 4.1 and conducted ablation studies to reflect real-world use cases of ModuLM. In these experiments, *w/o M-Encoder* refers to the exclusion of the customized encoder, *w/o Interaction* indicates the removal of the molecule interaction design, and *w/o CoT* represents the absence of the chain-of-thought reasoning mechanism. The full experimental results are reported in Table 6, while the outcomes of the ablation studies are summarized in Table 7.

Table 6: Performance of Custom Model on DDI and SSI Datasets

| Experiment | Accuracy (ChChMiner) | Accuracy (ZhangDDI) | Accuracy (DeepDDI) | RMSE (FreeSolv) | RMSE (CompSol) | RMSE (CombiSolv) |
|---|---|---|---|---|---|---|
| 1.7 | $0.953 \pm 0.010$ | $0.889 \pm 0.004$ | $0.942 \pm 0.007$ | $0.749 \pm 0.021$ | $0.415 \pm 0.019$ | $0.515 \pm 0.023$ |
| 1.8 | $0.964 \pm 0.005$ | $0.907 \pm 0.006$ | $0.959 \pm 0.010$ | $0.698 \pm 0.020$ | $0.298 \pm 0.018$ | $0.388 \pm 0.022$ |
| Custom Model | $\mathbf{0.968 \pm 0.006}$ | $\mathbf{0.911 \pm 0.006}$ | $\mathbf{0.964 \pm 0.008}$ | $\mathbf{0.680 \pm 0.019}$ | $\mathbf{0.288 \pm 0.013}$ | $\mathbf{0.359 \pm 0.013}$ |

Table 7: Results of the Ablation Study on DDI and SSI Datasets

| Experiment | Accuracy (ChChMiner) | Accuracy (ZhangDDI) | Accuracy (DeepDDI) | RMSE (FreeSolv) | RMSE (CompSol) | RMSE (CombiSolv) |
|---|---|---|---|---|---|---|
| w/o M-Encoder | $0.962 \pm 0.007$ | $0.905 \pm 0.005$ | $0.959 \pm 0.007$ | $0.700 \pm 0.019$ | $0.299 \pm 0.017$ | $0.370 \pm 0.020$ |
| w/o Interaction | $0.955 \pm 0.008$ | $0.896 \pm 0.006$ | $0.950 \pm 0.007$ | $0.705 \pm 0.018$ | $0.317 \pm 0.016$ | $0.388 \pm 0.019$ |
| w/o CoT | $0.959 \pm 0.006$ | $0.901 \pm 0.006$ | $0.956 \pm 0.008$ | $0.732 \pm 0.020$ | $0.335 \pm 0.018$ | $0.382 \pm 0.021$ |
| Full Model | $\mathbf{0.968 \pm 0.006}$ | $\mathbf{0.911 \pm 0.006}$ | $\mathbf{0.964 \pm 0.008}$ | $\mathbf{0.680 \pm 0.019}$ | $\mathbf{0.288 \pm 0.013}$ | $\mathbf{0.359 \pm 0.013}$ |

Through the above experiments, we validated ModuLM's strong capability in supporting user-defined models. Users can define custom encoders following our provided protocol and flexibly integrate them with other encoders and interaction layers as modular components. The customized model involves a more complex configuration, including user-defined blocks and additional operations such as stacking and flattening the outputs of ModuLM components. Moreover, with ModuLM's dynamic model construction mechanism, users can easily adjust the model structure to perform ablation studies. As shown in Table 6 both the CoT reasoning prompt and the interaction layer contribute to performance improvements. Additionally, our custom encoder module, which incorporates internal interaction mechanisms, also enhances model performance.

## 5   Conclusion

We propose ModuLM, a framework that supports the dynamic construction of LLM-based MRL models to address benchmarking challenges in LLM-driven molecular relational learning. ModuLM accommodates multiple molecular input formats and enables the flexible assembly of diverse model architectures, facilitating robust and scalable experimentation. The framework simplifies model development and standardizes the evaluation process across different architectures, ensuring fair and consistent benchmarking. By providing a flexible and modular platform, ModuLM not only advances research in the MRL field but also lays the foundation for cross-disciplinary collaboration and innovation, with broad applications in areas such as drug design and molecular interaction analysis, helping researchers accelerate critical biomedical discoveries.

**Limitations:** The benchmark experiments presented in this paper demonstrate the capabilities of the ModuLM framework in constructing and comparing various model architectures, yet they do not cover all possible model combinations. Our primary goal is to highlight ModuLM's advantages in enabling flexible model construction and efficient model comparison, showcasing its adaptability and effectiveness across diverse model architectures. While the current experiments provide valuable insights into the functionality and potential of the framework, a comprehensive exploration of all possible model combinations is beyond the scope of this study and is intended as a key direction for future research. We encourage the research community to further investigate the performance of different configurations using our framework, thus promoting the diversity and innovation of model architectures.

**Future Work:** In the future, we will continue to maintain and expand the components within the ModuLM framework to further enhance its flexibility and applicability. Specifically, we will focus on introducing additional types of encoders, interaction layers, and evaluation metrics, while also

expanding support for a broader range of LLMs to improve the generalizability and scalability of the framework. At the same time, we will deepen ModuLM's application in molecular relational learning tasks, particularly in areas such as drug discovery and protein interaction analysis. We expect that through continuous iteration and optimization, ModuLM will become a powerful tool for advancing molecular relational learning and interdisciplinary research, providing flexible and efficient technical support for future scientific endeavors.

## 6 Acknowledgement

This paper is partially supported by the National Natural Science Foundation of China (No.12227901). The AI-driven experiments, simulations and model training were performed on the robotic AI-Scientist platform of Chinese Academy of Sciences., Anhui Science Foundation for Distinguished Young Scholars (No.1908085J24), Natural Science Foundation of China (No.62502491).

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

# A Appendix

## A.1 General Information

### A.1.1 Links

The code for ModuLM is currently available in our code repository `https://github.com/ssjjjhw/ModuLM`. The repository provides reproduction scripts for the best-performing model, and if you wish to reproduce other experiments conducted in this paper, please refer to the Appendix A.4.

### A.1.2 Licenses

The ModuLM is under the CC BY 4.0. We, the authors, bear all responsibility in case of violation of rights.

## A.2 Details of Datasets

In ModuLM, we provide diverse datasets for DDI, SSI, and CSI tasks to comprehensively evaluate ModuLM's flexibility and generalizability. An overview of the datasets is provided in Table 8.

Table 8: Data statistics.

| Task | Dataset | $\mathcal{G}^1$ | $\mathcal{G}^2$ | Pairs |
|------|---------|------|------|-------|
| DDI | ZhangDDI | 548 | 548 | 48,548 |
| | ChChMiner | 1322 | 1322 | 48,514 |
| | DeepDDI | - | - | 192,284 |
| | TWOSIDES | 555 | 555 | 3,576,513 |
| SSI | MNSol | 372 | 86 | 2,275 |
| | FreeSolv | 560 | 1 | 560 |
| | CompSol | 442 | 259 | 3,548 |
| | Abraham | 1038 | 122 | 6,091 |
| | CombiSolv | 1,368 | 291 | 10,145 |
| | CombiSolv-QM | 11,029 | 284 | 1,000,000 |
| CSI | Chromophore | 7,016 | 365 | 20,236 |

**ZhangDDI** [79]. This dataset consists of 548 drugs with 48,548 labeled drug–drug interactions (DDIs). It integrates multiple drug similarity measures, such as chemical structure and target-based similarities, to enable similarity-aware DDI prediction. Due to its comprehensive annotation and focus on similarity information, ZhangDDI is widely used as a benchmark for evaluating similarity-based computational methods in drug interaction prediction.

**ChChMiner** [88]. ChChMiner provides a curated collection of 1,322 drugs and 48,514 interactions derived from FDA-approved labels and peer-reviewed literature. Emphasizing clinically validated DDIs, it serves as a reliable dataset for real-world risk assessment and pharmacovigilance studies. Its focus on verified clinical data makes it particularly valuable for safety-critical applications.

**DeepDDI** [56]. Comprising 192,284 labeled DDIs annotated with side effect profiles, DeepDDI leverages data from DrugBank to support multi-label classification of adverse drug interactions. This dataset enables models to predict not only the presence of interactions but also the specific adverse effects, thereby providing richer supervision for deep learning models in pharmacology.

**TWOSIDES** [63]. TWOSIDES features 555 drugs with over 3.5 million reported interactions encompassing 1,318 distinct interaction types. Extracted from FDA adverse event reporting systems, it offers a large-scale resource for investigating long-tail and rare DDI patterns. This dataset is instrumental for exploring subtle and complex interaction effects in post-market drug safety monitoring.

**Chromophore** [31]. This dataset includes 20,236 chromophore–solvent pairs with experimentally measured optical properties such as absorption wavelength, emission wavelength, and excited-state lifetime. Unreliable or outlier data entries have been rigorously filtered, and log-normalization is

applied to reduce skewness in target distributions. Chromophore enables in-depth studies of solvent effects on photophysical behavior at the molecular level.

**MNSol** [48]. MNSol contains 3,037 records of solvation or transfer free energies covering 790 solutes and 92 solvents. The dataset has been filtered to 2,275 high-quality solute–solvent combinations consistent with established benchmarks, providing a robust resource for training and evaluating solvation energy prediction models in computational chemistry.

**FreeSolv** [49]. FreeSolv offers 643 hydration free energy measurements of small molecules in aqueous solution, of which 560 experimental entries are retained after quality control. It serves as a standard benchmark for predicting aqueous solvation properties, critical for understanding solute–solvent interactions relevant to drug design and molecular biology.

**CompSol** [50]. CompSol focuses on 3,548 solute–solvent pairs emphasizing the role of hydrogen bonding in solvation energy. By targeting molecular interactions specifically related to hydrogen bonds, this dataset provides a fine-grained benchmark to evaluate models' ability to capture solvent effects driven by directional intermolecular forces.

**Abraham** [23]. Based on Abraham's linear solvation energy relationships, this dataset compiles 6,091 solute–solvent combinations from 1,038 solutes and 122 solvents. It supports solvent effect modeling through well-established physicochemical parameters, making it suitable for studies on quantitative structure–property relationships (QSPR) and solvent screening.

**CombiSolv** [69]. CombiSolv integrates four benchmark datasets — MNSol, FreeSolv, CompSol, and Abraham — into a comprehensive collection of 10,145 solute–solvent pairs. This unified dataset facilitates broader generalization and benchmarking of molecular solvation models across diverse chemical spaces and experimental conditions.

**CombiSolv-QM** [69]. CombiSolv-QM extends CombiSolv by including 1 million solute–solvent pairs generated via quantum mechanical (QM) simulations. This large-scale synthetic dataset enables the evaluation of model robustness and scalability, providing a valuable resource for training models that generalize well to chemically diverse and complex molecular systems.

### A.3    Details of ModuLM Components

### A.3.1    2D Graph Encoders

**GCN** [34]. Graph Convolutional Network introduces a spectral-based graph convolution operation by approximating localized filters with a normalized graph Laplacian. This enables efficient aggregation of neighboring node features, allowing semi-supervised learning directly on graph-structured data. GCN's simple yet powerful layer design laid foundational groundwork for many subsequent graph neural network models by balancing computational efficiency and expressiveness.

**MPNN** [22]. Message Passing Neural Network generalizes the idea of graph convolutions by explicitly learning both the message functions exchanged between nodes and the update functions that revise node states. This flexible framework supports incorporation of domain-specific knowledge, allowing improved molecular property predictions by capturing complex interactions through iterative message passing.

**GAT** [68]. Graph Attention Network innovatively applies self-attention mechanisms on graph neighborhoods, enabling the model to assign adaptive importance weights to different neighbors during embedding updates. This attention mechanism enhances the capacity to learn from irregular and heterogeneous graph structures, improving representation quality without requiring explicit graph preprocessing.

**NeuralFP** [13]. Neural Fingerprints pioneer end-to-end learning of molecular fingerprints using graph convolutional layers. Unlike handcrafted descriptors, this method automatically extracts relevant chemical features for property prediction tasks, showing superior performance and adaptability across diverse molecular datasets.

**AttentiveFP** [73]. Attentive Fingerprint leverages attention mechanisms at both atom and substructure levels, allowing the model to focus selectively on chemically important regions. This hierarchical attention not only improves prediction accuracy but also enhances interpretability, making it valuable for QSAR modeling and drug discovery.

**GIN** [74]. Graph Isomorphism Network achieves maximum discriminative power among GNNs by using injective neighborhood aggregation functions. This theoretical guarantee enables GIN to distinguish graph structures as effectively as the Weisfeiler-Lehman graph isomorphism test, pushing forward the limits of GNN expressiveness.

**GraphSAGE** [25]. Graph Sample and AggregatE introduces an inductive learning framework that samples and aggregates neighborhood features to generate node embeddings. This scalable approach enables GNNs to handle large, evolving graphs efficiently, supporting applications where graphs are dynamic or partially observed.

**CoATGIN** [80]. CoATGIN innovatively combines convolutional aggregation with attention mechanisms, capturing both local subgraph motifs and global graph contexts in molecular graphs. This hybrid approach enhances molecular representation quality by balancing locality and global awareness, beneficial for tasks requiring nuanced structural understanding.

### A.3.2   3D Conformation Encoders

**EGNN** [58]. Equivariant Graph Neural Network maintains equivariance to Euclidean transformations by jointly updating node features and their spatial coordinates. This design allows effective modeling of 3D molecular geometry without sacrificing rotational or translational invariance, crucial for accurate physical property predictions.

**3D-GeoFormer** [86]. 3D-GeoFormer employs geometric transformers to model spatial relationships in molecular structures, leveraging attention mechanisms adapted for 3D spatial data. This enables capturing long-range dependencies and complex geometric interactions, advancing 3D molecular representation learning beyond local neighborhoods.

**SE(3)-Transformer** [19]. SE(3)-Transformer integrates tensor field networks to ensure full SE(3) equivariance, allowing seamless handling of rotations and translations in 3D space. This makes it particularly suitable for modeling complex molecular and protein structures where geometric consistency under transformations is critical.

**PaiNN** [59]. PaiNN achieves rotational equivariance by explicitly separating scalar and vector features within its message-passing framework. This distinction enables precise modeling of directional physical quantities such as forces and energies, leading to state-of-the-art accuracy in quantum chemistry tasks.

**GVP** [30]. Geometric Vector Perceptrons combine scalar and vector features through specialized geometric transformations, enabling effective encoding of molecular structures in 3D. GVP generalizes across various 3D molecular graphs and shows strong performance on both structural and functional prediction tasks.

**GearNet** [81]. GearNet builds hierarchical molecular representations by integrating multi-scale structural and sequential information, capturing both local atomic details and broader biological context. This approach enhances model capability across different biological levels, supporting tasks like protein structure prediction and interaction analysis.

**DimeNet++** [21]. DimeNet++ improves directional message passing by explicitly encoding angular and directional information between atoms. Its architecture achieves higher accuracy and efficiency compared to its predecessor, effectively modeling molecular interactions that depend on precise geometric orientations.

**SchNet** [60]. SchNet introduces continuous-filter convolutional layers to model quantum interactions in molecular systems. Its end-to-end differentiable framework enables accurate predictions of atomic-level properties by capturing spatial correlations without requiring predefined features.

**SphereNet** [44]. SphereNet encodes spherical coordinates such as angles and radial distances to better capture 3D molecular geometry. This precise geometric encoding leads to improved performance on quantum property prediction tasks by more faithfully representing spatial relations.

**G-SphereNet** [46]. G-SphereNet extends SphereNet by incorporating an autoregressive generative process for molecular conformations, enabling both accurate representation and generation of complex 3D structures. This combination supports downstream applications in molecular design and simulation.

**Uni-Mol** [85]. Uni-Mol unifies pretraining and finetuning in 3D molecular representation learning via a transformer-based architecture. Its spatially aware design supports a wide variety of downstream tasks, improving generalization by leveraging large-scale molecular conformations during pretraining.

**GeoMFormer** [4]. GeoMFormer proposes a general transformer framework tailored to capture geometric relationships within molecular structures. By effectively modeling 3D spatial interactions, it improves the expressiveness and transferability of molecular representations across diverse tasks.

**GotenNet** [1]. GotenNet rethinks geometric message passing by designing an efficient 3D equivariant graph neural network that significantly reduces computational cost while maintaining or improving predictive accuracy. Its novel architectural choices enable scalable and effective modeling of complex molecular geometries.

### A.3.3 Interaction Layers

**Bilinear Attention** [2]. The Bilinear Attention Network (BAN) layer captures interactions between 2D feature sets through bilinear transformations, followed by attention pooling and batch normalization.

**Self Attention** [67]. Self-attention mechanisms allow a feature set to focus on its own elements, enabling models to capture relationships within the same source of data.

**Cross Attention** [54]. Cross-attention captures interactions between two distinct feature sets by applying attention mechanisms that focus on cross-source dependencies.

**Highway** [87]. The Highway mechanism combines 1D features through gated layers, allowing information to flow selectively by controlling the gates.

**Gated Fusion** [47]. Gated fusion combines 1D features from two sources by applying gated transformations, producing a unified representation that captures the interactions between them.

**Bilinear Fusion** [41]. Bilinear Fusion combines 1D features using a bilinear transformation and ReLU activation, capturing multiplicative interactions to enhance feature representation.

**Mean**. The Mean method combines feature sets by averaging their values.

### A.3.4 Backbones

In ModuLM, we provide seven popular LLMs: DeepSeek-1.5B, DeepSeek-7B, DeepSeek-14B [3], LLaMA-1B, LLaMA-13B [65], Galactica-1.3B, and Galactica-6.7B [64].

**DeepSeek.** The DeepSeek-1.5B, DeepSeek-7B, and DeepSeek-14B models are derived from the Qwen-2.5 series, which were originally licensed under the Apache 2.0 License, and have now been fine-tuned with 800k samples curated using DeepSeek-R1.

**LLama.** LLama 1B incorporated logits from the LLama 3.1 8B and 70B models during the pretraining stage, using the outputs (logits) from these larger models as token-level targets. Knowledge distillation was applied after pruning to recover performance. LLama-13B was pretrained on 2 trillion tokens of data from publicly available sources and fine-tuned on publicly available instruction datasets, along with over one million new human-annotated examples, making it a general-purpose LLM.

**Galactica.** Galactica-1.3B and Galactica-6.7B are large language models developed by Meta for scientific research and knowledge-intensive tasks. These models are designed to assist with tasks in fields like scientific literature, research summarization, and computational biology.

### A.3.5 Prompts

In the main text, we mentioned the two prompts involved in our experiments: Direct and Chain-of-Thought-based reasoning. The prompt design is shown in the Table 9.

### A.4 Example Usage of ModuLM

This section demonstrates how to construct example models from the main text using ModuLM. In ModuLM, we provide a configuration method based on a JSON file. In the modules we offer, users only need to modify the corresponding parameters. It is worth noting that some of the parameters

Table 9: Prompts settings.

---

**Direct Reasoning for Qualitative Tasks**
The first molecule `<SMILES0>` and the second molecule `<SMILES1>` are expected to interact with each other, potentially forming a molecular complex or influencing each other's properties.
**CoT-based Reasoning for Qualitative Tasks**
The first molecule `<SMILES0>` is likely to exhibit `[Property0]`, while the second molecule `<SMILES1>` is likely to exhibit `[Property1]`. Hence, the first drug molecule may alter the therapeutic effects of the second drug molecule. Therefore, they are likely to interact with each other.

---

**Direct Reasoning for Quantitative Tasks**
The solvation Gibbs free energy between the first molecule `<SMILES0>` and the second molecule `<SMILES1>` is 4.6232.
**CoT-based Reasoning for Quantitative Tasks**
The first molecule `<SMILES0>` is likely to exhibit `[Property0]`, while the second molecule `<SMILES1>` is likely to exhibit `[Property1]`. Therefore, their solvation Gibbs free energy is likely to fall between 4.0 and 4.5, with a precise value potentially being 4.6232.

---

below are included solely to demonstrate the comprehensiveness of our framework; if the goal is simply to use it, most hyperparameters do not need to be changed.

### A.4.1 Loading the Dataset

Taking the DeepDDI dataset loading as an example, the path to the dataset is defined using the `root` parameter. To accelerate data loading, the `num_workers` parameter is used to enable multi-threaded data processing. Additionally, the `use_3d` flag controls whether to incorporate 3D molecular conformational data as input. This allows users to flexibly switch between 2D and 3D molecular representations depending on the task requirements and available structural information.

```
{
    "root": "data/DDI/DeepDDI/",
    "num_workers": 5,
    "use_3d":true
}
```

### A.4.2 Initializing Encoder

In this setup, we utilize the Uni-Mol model as our 3D molecular conformation encoder. The specific encoder is selected by setting the `graph3d` parameter accordingly. To fine-tune its behavior and architecture, we provide a dedicated configuration file that defines key hyperparameters such as the number of layers, embedding dimensions, attention mechanisms, and dropout rates. This modular design allows for flexible customization and seamless integration into various molecular representation learning tasks.

```
{
    "graph3d": "unimol",
    "con_activation_dropout": 0.0,
    "con_activation_fn": "gelu",
    "con_attention_dropout": 0.1,
    "con_delta_pair_repr_norm_loss": -1.0,
    "con_dropout": 0.1,
    "con_emb_dropout": 0.1,
    "con_encoder_attention_heads": 64,
    "con_encoder_embed_dim": 512,
    "con_encoder_ffn_embed_dim": 2048,
    "con_encoder_layers": 15,
    "con_max_atoms": 256,
    "con_max_seq_len": 512
}
```

It is important to note that in order to build the example model provided in the main text, we need to make further modifications and extensions to the Uni-Mol model. Once the code has been extended, we can simply place it in the specified directory and continue managing and calling it through the config file.

### A.4.3 Configuring LLM

Here, the `mode` specifies the model's mode, whether it is pretraining or fine-tuning. The backbone parameter indicates the LLM to be used, while `min_len` and `max_len` define the minimum and maximum lengths of the generated text. Additional details and parameters for LLM-based text generation are provided in our code repository; this section highlights only a few key settings as examples.

```
{    "mode":"ft",
     "backbone": "DeepSeek-1.5B",
     "min_len": 10,
     "max_len": 40
}
```

### A.4.4 Training the Model

After constructing the model, we can fine-tune it by configuring the appropriate LoRA file. To make it easier for others to fine-tune using our framework, we provide the LoRA parameter configuration for model training here.

```
{
     "base_model_name_or_path": null,
     "bias": "none",
     "fan_in_fan_out": false,
     "inference_mode": false,
     "init_lora_weights": true,
     "lora_alpha": 32,
     "lora_dropout": 0.1,
     "target_modules": ["q_proj", "v_proj", "out_proj", "fc1", "fc2"],
     "peft_type": "LORA",
     "r": 16,
     "modules_to_save": null,
     "task_type": "CAUSAL_LM"
}
```

The specific batch size and number of training epochs are also configured in a unified manner. Here, the `batch_size` specifies the number of samples processed in each training step, which in this case is set to 12. The `max_epochs` defines the total number of training iterations over the entire dataset, set here to 20 epochs. The `save_every_n_epochs` parameter indicates that the model's state will be saved every 5 epochs. The `scheduler` field specifies the learning rate scheduling strategy—`linear_warmup_cosine_lr` gradually increases the learning rate during a warm-up period, then decays it following a cosine curve. The `seed` ensures reproducibility of training results by fixing randomness. `warmup_lr` and `warmup_steps` define the initial learning rate and the number of steps over which it will warm up, respectively. Lastly, `weight_decay` is used as a regularization technique to prevent overfitting by penalizing large weights during optimization.

```
{
     "batch_size": 12,
     "max_epochs": "30",
     "save_every_n_epochs": 5,
     "scheduler": linear_warmup_cosine_lr,
     "seed": 42,
     "warmup_lr": 1e-06,
     "warmup_steps": 1000,
     "weight_decay": "0.05"
}
```

### A.5 More experimental details

#### A.5.1 Details of Experimental Setup

In this section, we provide a more detailed description of the experimental data and testing configurations used in the main text.

**Training Epochs.** At the beginning of each experiment, we initiate incremental pretraining by running 5 epochs on the collected pretraining dataset. During the subsequent fine-tuning stage, the number of training epochs is task-dependent. Specifically, for the DDI task, we fine-tune the model for 50 epochs. For SSI datasets containing more than 3000 molecular pairs, we adopt a two-stage fine-tuning strategy: first, the model is fine-tuned on the CombiSolv-QM dataset for 100 epochs, followed by an additional 30 epochs on the target dataset. In contrast, for SSI datasets with fewer than 3000 molecular pairs, the fine-tuning stage is shortened to 20 epochs. Notably, both pretraining and fine-tuning phases share the same optimizer and learning rate scheduling configurations, as described in the following section. It is worth noting that during training, we did not employ an early stopping strategy. Instead, we saved the model's performance on both the validation and test sets after each epoch to facilitate more thorough analysis. When reporting the results, we selected the test set performance corresponding to the epoch with the best validation set performance for evaluation.

**Training Strategy.** We employ the AdamW optimizer with a weight decay coefficient of 0.05 to mitigate overfitting and stabilize training. The learning rate is governed by a linear warm-up followed by cosine decay schedule, which helps accelerate convergence in the early stages and enables refined optimization during the later phases. To further enhance efficiency and reduce training overhead, we adopt Low-Rank Adaptation (LoRA), implemented using the OpenDelta and PEFT libraries. The rank parameter of LoRA is set to $r = 16$. For models in the DeepSeek series, LoRA is applied to the following modules: [q_proj, k_proj, v_proj, o_proj, gate_proj, up_proj, down_proj]. For the LLaMA and Galactica models, LoRA is instead integrated into [q_proj, v_proj, out_proj, fc1, fc2].

#### A.5.2 The Impact of Post Pretraining

Table 10: Performance of different Post Pretraining Stragedy.

| Experiment | Accuracy (ChChMiner) | Accuracy (ZhangDDI) | Accuracy (DeepDDI) | RMSE (FreeSolv) | RMSE (CompSol) | RMSE (CombiSolv) |
|---|---|---|---|---|---|---|
| Molecular Interaction-based | $0.962 \pm 0.007$ | $0.911 \pm 0.006$ | $0.954 \pm 0.008$ | $0.682 \pm 0.017$ | $0.301 \pm 0.016$ | $0.374 \pm 0.019$ |
| Substructure-based | $0.961 \pm 0.007$ | $0.908 \pm 0.006$ | $0.956 \pm 0.007$ | $0.692 \pm 0.017$ | $0.295 \pm 0.017$ | $0.372 \pm 0.019$ |
| Structure Similarity-guided | $0.966 \pm 0.004$ | $0.914 \pm 0.005$ | $0.961 \pm 0.008$ | $0.676 \pm 0.020$ | $0.292 \pm 0.018$ | $0.367 \pm 0.022$ |

In the main text, we integrated three incremental pretraining methods to help the model better acquire domain knowledge relevant to molecular relational learning. To evaluate the impact of these pretraining strategies on model performance, we conducted a series of tests. Specifically, we selected the best-performing model from the main text as the evaluation baseline. Since conducting zero-shot evaluations using pure textual input can lead to unreliable or unreportable results on certain datasets, we opted to retain the fine-tuning process and only replace the pretraining strategy during testing. It is important to note that our experiments in the main text have already demonstrated that DeepSeek consistently outperforms other models under the same pretraining settings. Therefore, we conduct all evaluations here using the DeepSeek-1.5B model. The results are shown in Table 10.

#### A.5.3 Specialized Encoder Analysis

To better compare the performance differences introduced by various encoders, we standardize all other configurations and only replace the encoder components for analysis. The experimental setup follows a similar design to that used in the main body, employing an MLP for modality alignment and using DeepSeek-1.5B as the backbone LLM. It is important to emphasize that the main contribution of this work is to provide a fair, flexible, and extensible benchmark framework for MRL. Given the combinatorial complexity of possible encoder and architecture choices, our focus is on evaluation rather than theoretical analysis. We believe that performance differences stemming from specific encoders or architectures should be thoroughly analyzed and justified by the authors of the respective methods. As mentioned in the main text, 3D molecular conformations contain richer structural information; therefore, to aid readers in gaining a deeper understanding of the MRL task, we provide

a brief comparative discussion of Uni-Mol and GotenNet as representative examples to highlight structural design differences and their potential impact on performance. Overall performance is shown in Table 11.

Table 11: Overall Performance of different Encoders.

| Experiment | Accuracy (ChChMiner) | Accuracy (ZhangDDI) | Accuracy (DeepDDI) | RMSE (FreeSolv) | RMSE (CompSol) | RMSE (CombiSolv) |
|---|---|---|---|---|---|---|
| Uni-Mol | $0.959 \pm 0.010$ | $0.903 \pm 0.004$ | $0.952 \pm 0.007$ | $0.690 \pm 0.021$ | $0.310 \pm 0.019$ | $0.387 \pm 0.023$ |
| GotenNet | $0.966 \pm 0.004$ | $0.914 \pm 0.005$ | $0.961 \pm 0.008$ | $0.676 \pm 0.020$ | $0.292 \pm 0.018$ | $0.367 \pm 0.022$ |

GotenNet outperforms Uni-Mol primarily due to its innovative design in modeling three-dimensional geometric information. Firstly, GotenNet employs an efficient and refined geometric message-passing mechanism that captures spatial relationships and angular dependencies within molecules more accurately, enabling a more comprehensive understanding of molecular structural complexity. Secondly, while maintaining SE(3) equivariance, GotenNet optimizes computational efficiency by reducing redundant calculations, thereby improving both training and inference speed without compromising representational capacity. Moreover, GotenNet places greater emphasis on multi-scale feature integration, combining local details with global conformational information, which enhances its generalization ability across various molecular property prediction tasks.

In contrast, although Uni-Mol also utilizes a transformer-based 3D encoding strategy, its handling of geometric information is comparatively coarser and incurs higher computational costs, limiting the model's scalability and performance improvements. Therefore, through structural innovations and efficiency optimizations, Goten-Net achieves a deeper understanding of 3D molecular information, leading to superior performance. The detailed computational efficiency can be found in Table 12. Here, we use the average number of samples processed per second on a single GPU as the evaluation metric.

Table 12: Efficiency of different encoders.

| **Model** | Uni-Mol | GotenNet |
|---|---|---|
| **Rate** | 2.74 it/s | 2.66 it/s |

Table 13: Performance of different encoders on molecules of varying sizes.

| Experiment | 344 | 566 | 628 | 730 | 1846 |
|---|---|---|---|---|---|
| Uni-Mol | $0.879 \pm 0.012$ | $0.863 \pm 0.014$ | $0.852 \pm 0.019$ | $0.833 \pm 0.020$ | $0.739 \pm 0.022$ |
| GotenNet | $0.891 \pm 0.010$ | $0.884 \pm 0.013$ | $0.873 \pm 0.018$ | $0.845 \pm 0.019$ | $0.752 \pm 0.021$ |

To further validate whether the performance differences between GotenNet and Uni-Mol across molecules of varying sizes align with their respective architectural designs, we conducted additional experiments. We combined ZhangDDI, ChChDDI, DeepDDI, and TWOSIDES into a large aggregated dataset, grouping the samples by molecular mass with 20,000 samples per group. Here, AM denotes the average molecular mass within each group. Accuracy was used as the evaluation metric. For detailed performance results, please refer to the Table 13.

### A.5.4 Computational Efficiency Analysis

To better compare the differences in computational efficiency brought by various backbones, encoding methods, and encoders, we conducted a more systematic time efficiency comparison following a similar approach as described above. The detailed results are shown in Table 15. To better highlight the differences, we selectively tested a subset of configurations presented in the main text. The details are shown in Table 14.

From the Table 15, we can observe that the primary factor affecting computational efficiency across different model configurations is the choice of backbone. Larger backbones significantly increase the computational cost. The second key factor is modality fusion—integrating more modalities increases processing complexity, thereby reducing efficiency. Notably, when using Q-former for alignment, the performance overhead becomes especially pronounced due to the large number of trainable parameters typically involved. For example, in configuration 1.16*, where all modalities are fused and Q-former is used for alignment, the processing rate drops to a very low level. Therefore, this

Table 14: Experimental Settings on DDI Datasets

| Experiment No. | Backbone | Encoder | Interaction | Input Feature |
|---|---|---|---|---|
| 1.2 | Galactica-1.3B | GIN | - | $m_s + m_g$ |
| 1.3 | Galactica-1.3B | GIN | Cross Attention | $m_s + m_g$ |
| 1.6 | DeepSeek-1.5B | - | - | $m_s$ |
| 1.7 | DeepSeek-1.5B | GIN | - | $m_s + m_g$ |
| 1.8 | DeepSeek-1.5B | GotenNet | - | $m_s + m_c$ |
| 1.10 | DeepSeek-14B | GotenNet | - | $m_s + m_c$ |
| 1.16 | DeepSeek-1.5B | GIN+GotenNet | - | $m_s + m_g + m_c$ |

**Note:** $m_s$ = molecular sequence, $m_g$ = molecular graph, $m_c$ = molecular conformation. '-' indicates that no method is applied.

Table 15: Efficiency of different encoders. * indicates the use of Q-former for alignment.

| Model | 1.2 | 1.3 | 1.6 | 1.7 | 1.7* | 1.8 | 1.8* | 1.10 | 1.16 | 1.16* |
|---|---|---|---|---|---|---|---|---|---|---|
| Rate | 3.16 it/s | 3.10 it/s | 4.22 it/s | 3.38 it/s | 2.98 it/s | 2.66 it/s | 2.28 it/s | 1.22 it/s | 2.01 it/s | 1.11 it/s |

table is intended to help users of our framework make informed decisions about model configurations, enabling a balanced trade-off between computational efficiency and model performance.

### A.5.5 Impact of Molecular Dataset

Meanwhile, to facilitate the exploration of model adaptability across different dataset scales and molecular sizes under various configurations, we first conducted experiments by grouping molecules based on their sizes, following a similar approach as described above. Additionally, we performed experiments using sampled subsets of varying scales from the ZhangDDI dataset for comparative analysis. Detailed experimental results can be found in Tables 16 and 17.

Table 16: Model performance variation with respect to molecular size, using Accuracy as the evaluation metric.

| Model | 1.2 | 1.3 | 1.6 | 1.7 | 1.7* | 1.8 | 1.8* | 1.10 | 1.16 |
|---|---|---|---|---|---|---|---|---|---|
| 334 | $0.868_{(.014)}$ | $0.874_{(.012)}$ | $0.855_{(.019)}$ | $0.879_{(.017)}$ | $0.881_{(.015)}$ | $0.892_{(.014)}$ | $0.890_{(.015)}$ | $0.860_{(.013)}$ | $0.851_{(.017)}$ |
| 566 | $0.851_{(.015)}$ | $0.863_{(.014)}$ | $0.843_{(.018)}$ | $0.864_{(.016)}$ | $0.869_{(.015)}$ | $0.872_{(.016)}$ | $0.865_{(.017)}$ | $0.845_{(.014)}$ | $0.837_{(.016)}$ |
| 628 | $0.825_{(.016)}$ | $0.842_{(.015)}$ | $0.818_{(.017)}$ | $0.838_{(.018)}$ | $0.840_{(.016)}$ | $0.845_{(.015)}$ | $0.832_{(.016)}$ | $0.820_{(.015)}$ | $0.812_{(.017)}$ |
| 730 | $0.790_{(.018)}$ | $0.804_{(.017)}$ | $0.780_{(.019)}$ | $0.812_{(.017)}$ | $0.815_{(.014)}$ | $0.822_{(.013)}$ | $0.809_{(.014)}$ | $0.785_{(.016)}$ | $0.778_{(.018)}$ |
| 1846 | $0.701_{(.019)}$ | $0.714_{(.017)}$ | $0.687_{(.020)}$ | $0.721_{(.011)}$ | $0.717_{(.012)}$ | $0.754_{(.010)}$ | $0.747_{(.009)}$ | $0.690_{(.019)}$ | $0.742_{(.016)}$ |

The data presented in the table clearly indicate that, regardless of the model combinations employed, the capability to process large molecules falls short of expectations. This limitation is not unique to our approach; it is a common challenge observed across existing multimodal molecular representation models. The sheer size of large molecules not only leads to a significant decrease in computational efficiency but also introduces an overwhelming amount of complex information. Current encoding models struggle to effectively manage this complexity. In the context of LLMs, our methodology involves simply concatenating the molecular text modality with its corresponding multimodal data before feeding it into the LLM for inference. This approach aligns with the prevailing standard for multimodal LLMs. However, when the multimodal data cannot be encoded effectively, it may actually disrupt or confuse the LLM's decision-making process. Additionally, the excessively long molecular expressions associated with large molecules themselves pose further challenges, negatively impacting the LLM's ability to generate accurate predictions. Overall, these factors highlight fundamental bottlenecks in scaling current models to handle large molecular structures effectively.

Due to the limitations of multimodal models, we are unable to perform zero-shot testing on them, which would lead to a complete collapse in model performance. Based on the data above, we observe that when training with LLMs, an insufficient amount of data often prevents the model from learning meaningful representations. This issue becomes even more pronounced in multimodal LLMs. Without exposing the model to enough diverse molecular examples from different modalities during training, the LLM is unable to accurately understand or infer the underlying semantics of the encoded information. In our experiments, we found that when using a zero-shot setting—i.e.,

Table 17: Model performance variation with respect to datasets size, using Accuracy as the evaluation metric.

| Size | 1.2 | 1.3 | 1.6 | 1.7 | 1.7* | 1.8 | 1.8* | 1.10 | 1.16 |
|---|---|---|---|---|---|---|---|---|---|
| **1000** | $0.645_{(.012)}$ | $0.650_{(.013)}$ | $0.641_{(.012)}$ | $0.673_{(.011)}$ | $0.675_{(.015)}$ | $0.703_{(.012)}$ | $0.687_{(.015)}$ | $0.681_{(.014)}$ | $0.697_{(.014)}$ |
| **5000** | $0.698_{(.013)}$ | $0.704_{(.013)}$ | $0.687_{(.014)}$ | $0.721_{(.012)}$ | $0.726_{(.014)}$ | $0.745_{(.013)}$ | $0.735_{(.014)}$ | $0.728_{(.013)}$ | $0.742_{(.012)}$ |
| **10000** | $0.772_{(.014)}$ | $0.779_{(.012)}$ | $0.761_{(.013)}$ | $0.793_{(.012)}$ | $0.799_{(.014)}$ | $0.818_{(.013)}$ | $0.808_{(.013)}$ | $0.802_{(.012)}$ | $0.815_{(.013)}$ |
| **20000** | $0.796_{(.014)}$ | $0.802_{(.012)}$ | $0.786_{(.014)}$ | $0.817_{(.012)}$ | $0.821_{(.013)}$ | $0.842_{(.012)}$ | $0.835_{(.013)}$ | $0.828_{(.012)}$ | $0.840_{(.012)}$ |
| **50000** | $0.866_{(.017)}$ | $0.870_{(.014)}$ | $0.841_{(.009)}$ | $0.886_{(.017)}$ | $0.881_{(.015)}$ | $0.902_{(.013)}$ | $0.897_{(.017)}$ | $0.860_{(.013)}$ | $0.892_{(.012)}$ |

directly testing the model without any prior training on our specific data—the LLM occasionally produced incoherent or grammatically incorrect outputs, indicating a lack of grounding in the input structure. Such behavior is clearly undesirable. However, as shown in the table, when the training data size is controlled and kept roughly consistent across settings, the relative ranking of model configurations in terms of performance remains largely unchanged. This suggests that the key differences between models lie not only in how much data they are exposed to, but also in how effectively their encoders are able to communicate molecular information to the LLM. In other words, a model's ability to maintain performance across varying data sizes reflects the encoder's competence in shaping representations that are both interpretable and informative to the LLM.

### A.5.6   More Experimental Results

Due to space limitations in the main text, we present additional experimental results here, with the experimental setup consistent with the one described in the main text.

Table 18: More Results of DDI Datasets

| Experiment | AUC-ROC (Drugbank) | Accuracy (Drugbank) | AUC-ROC (TWOSIDES) | Accuracy (TWOSIDES) |
|---|---|---|---|---|
| Chem T5[7] | $0.921 \pm 0.010$ | $0.859 \pm 0.013$ | $0.906 \pm 0.015$ | $0.856 \pm 0.022$ |
| MolCA[45] | $0.934 \pm 0.018$ | $0.898 \pm 0.010$ | $0.942 \pm 0.014$ | $0.907 \pm 0.015$ |
| MolT5[15] | $0.930 \pm 0.016$ | $0.904 \pm 0.018$ | $0.940 \pm 0.013$ | $0.929 \pm 0.017$ |
| MolTC[17] | $0.978 \pm 0.006$ | $0.951 \pm 0.005$ | $0.980 \pm 0.005$ | $0.970 \pm 0.007$ |
| 1.1 | $0.933 \pm 0.011$ | $0.891 \pm 0.012$ | $0.912 \pm 0.017$ | $0.877 \pm 0.014$ |
| 1.2 | $0.945 \pm 0.010$ | $0.922 \pm 0.011$ | $0.957 \pm 0.009$ | $0.923 \pm 0.009$ |
| 1.3 | $0.950 \pm 0.009$ | $0.935 \pm 0.010$ | $0.950 \pm 0.008$ | $0.926 \pm 0.008$ |
| 1.4 | $0.955 \pm 0.008$ | $0.938 \pm 0.009$ | $0.946 \pm 0.007$ | $0.935 \pm 0.008$ |
| 1.5 | $0.946 \pm 0.010$ | $0.931 \pm 0.010$ | $0.951 \pm 0.008$ | $0.918 \pm 0.008$ |
| 1.6 | $0.938 \pm 0.016$ | $0.901 \pm 0.012$ | $0.920 \pm 0.017$ | $0.893 \pm 0.018$ |
| 1.7 | $0.963 \pm 0.007$ | $0.944 \pm 0.007$ | $0.970 \pm 0.006$ | $0.952 \pm 0.006$ |
| 1.8 | $0.975 \pm 0.006$ | $0.950 \pm 0.006$ | $0.982 \pm 0.005$ | $0.975 \pm 0.005$ |
| 1.9 | $0.950 \pm 0.008$ | $0.937 \pm 0.008$ | $0.971 \pm 0.006$ | $0.949 \pm 0.006$ |
| 1.10 | $0.940 \pm 0.010$ | $0.926 \pm 0.010$ | $0.961 \pm 0.007$ | $0.938 \pm 0.007$ |
| 1.11 | $0.920 \pm 0.014$ | $0.886 \pm 0.010$ | $0.907 \pm 0.021$ | $0.855 \pm 0.020$ |
| 1.12 | $0.935 \pm 0.011$ | $0.920 \pm 0.010$ | $0.945 \pm 0.012$ | $0.911 \pm 0.008$ |
| 1.13 | $0.947 \pm 0.008$ | $0.933 \pm 0.009$ | $0.953 \pm 0.013$ | $0.923 \pm 0.009$ |
| 1.14 | $0.952 \pm 0.007$ | $0.931 \pm 0.009$ | $0.959 \pm 0.012$ | $0.932 \pm 0.010$ |
| 1.15 | $0.935 \pm 0.019$ | $0.877 \pm 0.020$ | $0.927 \pm 0.010$ | $0.894 \pm 0.011$ |
| 1.16 | $0.971 \pm 0.008$ | $0.948 \pm 0.004$ | $0.984 \pm 0.007$ | $0.978 \pm 0.003$ |
| Custom Model | $\mathbf{0.982 \pm 0.010}$ | $\mathbf{0.956 \pm 0.007}$ | $\mathbf{0.986 \pm 0.007}$ | $\mathbf{0.980 \pm 0.009}$ |

The results presented in Table 19 and Table 18 are largely consistent with those reported in the main text, reaffirming the trends observed across different configurations. Notably, the introduction of additional modality information consistently yields substantial improvements in model performance, highlighting the effectiveness of leveraging multimodal signals in enhancing representation learning and generalization. In contrast, architectural modifications that increase model complexity—such

Table 19: More Results of SSI Datasets

| Experiment | MAE (MNSol) | RMSE (MNSol) | MAE (Abraham) | RMSE (Abraham) |
|---|---|---|---|---|
| Chem T5[7] | $0.537 \pm 0.092$ | $1.011 \pm 0.083$ | $0.621 \pm 0.027$ | $0.918 \pm 0.032$ |
| MolCA[45] | $0.511 \pm 0.034$ | $0.956 \pm 0.049$ | $0.580 \pm 0.026$ | $0.910 \pm 0.032$ |
| MolT5[15] | $0.466 \pm 0.067$ | $0.867 \pm 0.069$ | $0.544 \pm 0.028$ | $0.833 \pm 0.029$ |
| MolTC[17] | $0.354 \pm 0.018$ | $0.625 \pm 0.023$ | $0.211 \pm 0.018$ | $0.390 \pm 0.021$ |
| 1.1 | $0.510 \pm 0.051$ | $0.971 \pm 0.063$ | $0.572 \pm 0.024$ | $0.865 \pm 0.030$ |
| 1.2 | $0.451 \pm 0.042$ | $0.890 \pm 0.028$ | $0.524 \pm 0.022$ | $0.813 \pm 0.027$ |
| 1.3 | $0.436 \pm 0.018$ | $0.877 \pm 0.027$ | $0.496 \pm 0.024$ | $0.804 \pm 0.029$ |
| 1.4 | $0.410 \pm 0.017$ | $0.808 \pm 0.024$ | $0.447 \pm 0.020$ | $0.681 \pm 0.023$ |
| 1.5 | $0.506 \pm 0.036$ | $0.944 \pm 0.043$ | $0.522 \pm 0.030$ | $0.787 \pm 0.031$ |
| 1.6 | $0.502 \pm 0.045$ | $0.936 \pm 0.052$ | $0.591 \pm 0.028$ | $0.863 \pm 0.025$ |
| 1.7 | $0.386 \pm 0.034$ | $0.727 \pm 0.035$ | $0.394 \pm 0.026$ | $0.512 \pm 0.029$ |
| 1.8 | $0.343 \pm 0.017$ | $0.618 \pm 0.024$ | $0.204 \pm 0.017$ | $0.408 \pm 0.021$ |
| 1.9 | $0.488 \pm 0.029$ | $0.834 \pm 0.031$ | $0.416 \pm 0.027$ | $0.562 \pm 0.029$ |
| 1.10 | $0.501 \pm 0.030$ | $0.851 \pm 0.032$ | $0.456 \pm 0.018$ | $0.618 \pm 0.022$ |
| 1.11 | $0.499 \pm 0.016$ | $0.859 \pm 0.023$ | $0.437 \pm 0.021$ | $0.620 \pm 0.030$ |
| 1.12 | $0.550 \pm 0.047$ | $1.118 \pm 0.064$ | $0.614 \pm 0.029$ | $0.933 \pm 0.030$ |
| 1.13 | $0.475 \pm 0.036$ | $0.878 \pm 0.033$ | $0.511 \pm 0.019$ | $0.764 \pm 0.021$ |
| 1.14 | $0.461 \pm 0.021$ | $0.818 \pm 0.027$ | $0.489 \pm 0.020$ | $0.713 \pm 0.021$ |
| 1.15 | $0.597 \pm 0.031$ | $0.864 \pm 0.035$ | $0.422 \pm 0.021$ | $0.683 \pm 0.023$ |
| 1.16 | $0.352 \pm 0.020$ | $0.625 \pm 0.018$ | $0.201 \pm 0.015$ | $0.389 \pm 0.017$ |
| Custom Model | $\mathbf{0.340 \pm 0.011}$ | $\mathbf{0.601 \pm 0.010}$ | $\mathbf{0.199 \pm 0.008}$ | $\mathbf{0.379 \pm 0.011}$ |

as altering internal modules or adding more parameters—do lead to moderate performance gains. However, these improvements are generally less pronounced compared to those achieved through the integration of new modalities. This suggests that the diversity and complementarity of multimodal data play a more critical role than mere architectural sophistication in driving performance gains.

In addition to the DDI and SSI datasets, we further evaluate our framework on the CSI dataset to demonstrate its comprehensiveness and scalability. It is worth noting that the performance of models under different configurations on the CSI dataset varies significantly. To facilitate a more intuitive comparison of the performance across different settings, we report the best-performing configuration for each backbone. The results are summarized in the Table 20. Note that the three datasets in the CSI domain are all derived by splitting the Chromophore dataset.

Table 20: Performance on CSI Datasets

| Experiment | MAE (Absorption) | RMSE (Absorption) | MAE (Emission) | RMSE (Emission) | MAE (Lifetime) | RMSE (Lifetime) |
|---|---|---|---|---|---|---|
| MolTC[17] | $17.55 \pm 1.83$ | $29.10 \pm 2.15$ | $20.22 \pm 1.91$ | $34.17 \pm 2.02$ | $0.911 \pm 0.052$ | $1.213 \pm 0.092$ |
| 1.4 | $18.67 \pm 2.01$ | $31.33 \pm 2.47$ | $22.37 \pm 2.01$ | $36.71 \pm 2.93$ | $1.011 \pm 0.061$ | $1.502 \pm 0.103$ |
| 1.8 | $16.71 \pm 1.82$ | $28.67 \pm 1.99$ | $19.08 \pm 1.89$ | $38.00 \pm 1.87$ | $0.943 \pm 0.054$ | $\mathbf{1.119 \pm 0.077}$ |
| 1.14 | $19.01 \pm 2.01$ | $32.46 \pm 2.92$ | $21.84 \pm 1.96$ | $37.56 \pm 3.02$ | $1.009 \pm 0.060$ | $1.711 \pm 0.095$ |
| Custom Model | $\mathbf{15.42 \pm 1.53}$ | $\mathbf{27.65 \pm 1.91}$ | $\mathbf{17.11 \pm 1.68}$ | $\mathbf{31.55 \pm 1.60}$ | $\mathbf{0.929 \pm 0.064}$ | $1.123 \pm 0.082$ |

From the data in the Table above, we can clearly analyze the performance differences of various models under different experimental settings. Leveraging the usability and extensibility of ModuLM, we can implement and compare a wider range of LLM-based MRL models, thereby gaining insights into how model design impacts performance.

