# OpenReview forum: "ModuLM: Enabling Modular and Multimodal Molecular Relational Learning with Large Language Models"
_NeurIPS.cc/2025/Datasets_and_Benchmarks_Track — NeurIPS 2025 Datasets and Benchmarks Track poster_

### Official Review · Reviewer_Kinq · 2025-06-21

**Rating:** 6
**Confidence:** 4

**Summary:**

This paper proposes a modular LLM framework for MRL, addressing the challenges of complex module extension and difficulty in supporting different modalities in the field of LLM-based MRL. It also conducts extensive experiments involving multiple modalities, multiple encoders, and various backbones. The paper is clearly structured, so I believe this paper deserves to be accepted.

**Dataset Code Accessibility:**

Yes

**Dataset Code Comments:**

The paper provides the code repository address, which includes the sources of the datasets used in the paper and corresponding scripts for running tests. Regarding the experiments conducted in the paper, the repository also offers running examples.

**Ethical Comments:**

After careful evaluation, this paper complies with relevant laws, regulations, and ethical standards in data collection, usage, and processing. The datasets used have been legally authorized or publicly released and do not involve sensitive personal information or privacy risks. The methods and experiments proposed in the paper do not raise obvious issues of bias or discrimination, and the authors have taken a cautious approach to societal impact. Overall, the authors have given full consideration to ethical factors, and no significant ethical risks or potential negative societal impacts have been identified.

**Ethical Considerations:**

No, there are no or only very minor ethics concerns

**Final Justification:**

Thank you for the author's response. The concerns I raised have been adequately addressed. This work represents the first fair and comprehensive benchmarking framework for molecular relation learning tasks based on LLMs. Upon further reflection, I have decided to raise my score.

**Limitations Weaknesses:**

Disadvantages:
The paper does not discuss how potential conflicts arising from custom module extensions are addressed in Section 4.3, Custom Model Design and Evaluation.

**Strengths Contributions:**

Clarity and Organization: The introduction, methodology, and experiments sections of the paper are clearly written and logically structured, helping readers easily understand the motivation, methodology, and conclusions.

Novel Framework: The authors propose the ModuLM framework, which adopts a modular approach to evaluating Molecular Relation Learning models based on LLMs. This design addresses the pressing need for systematic and extensible evaluation tools in this domain.

Flexibility and Multi-Modality: The ModuLM framework is easy to configure and supports multi-modal inputs, making it applicable to a wide range of molecular data scenarios. This flexibility enhances its practical value and potential for future expansion.

Thorough Empirical Validation: In experiment, the paper conducts extensive empirical studies involving different modalities, encoders, and various LLM backbones, effectively filling the gap in systematic evaluation of LLMs within the MRL field.

---

> ### Author Rebuttal · Authors · 2025-07-28
>
> Thank you for your detailed and positive review of our paper. We sincerely appreciate your recognition of the paper's clarity, organization, and originality. Your affirmation of the ModuLM framework’s modular design, flexibility, and thorough experimental validation is truly encouraging for us. Below, we will carefully address your questions.
>
>
> > **W1.** *The paper does not discuss how potential conflicts arising from custom module extensions are addressed.*
>
> Thank you for your feedback. If you wish to integrate custom modules into the existing model framework, you only need to ensure that the input and output interfaces are properly aligned. ModuLM already provides preprocessed data and standardized data loading methods. Under this unified format, there will be no conflict issues among internal modules within ModuLM. Here, we provide a detailed list of dimensions for reference.
> Here, we follow a structured process: **Aft_2D_Encoder** refers to the dimensions that need to be ensured after encoding the molecular graph; **Aft_3D_Encoder** refers to the dimensions required after encoding the molecular conformation; **Pre_Alignment** denotes the dimensions that must be maintained before alignment; and **Pre_LLM** indicates the dimensions that need to be ensured before being input into the LLM. We have adopted a unified configuration here to facilitate future extension and direct integration. If more flexibility regarding dimensional variations is needed, we will provide a separate, modular configuration in future work to make adjustments more convenient.
>
> | **Aft_2D_Encoder** | **Aft_3D_Encoder** | **Pre_Alignment** | **Pre_LLM** |
> | ------------------ | ------------------ | ----------------- | ----------- |
> | 1024               | 1024               | 2048              | 2048        |
>
> As for potential dependency conflicts that may arise from other custom encoders used in the framework, we do not elaborate on them further in this paper. During our experiments, we integrated a variety of recently developed encoders, and no version conflicts occurred when installing dependencies based on our specified environment. If any environment-related issues arise while incorporating custom encoding modules, we welcome further discussion and are open to providing support.
>
> ---------------
>
> We sincerely thank you for your positive evaluation of our manuscript. Your recognition has been truly encouraging and has further motivated us to continue our exploration in the field of MRL. We hope that our responses have sufficiently addressed your concerns and clarified any ambiguities. If so, we would respectfully appreciate your consideration of a higher score. Should you have any further questions or require additional clarification, we would be more than willing to engage in further discussion. Once again, we are deeply grateful for the time and effort you have dedicated to reviewing our work. Your valuable feedback has played an important role in improving the quality of our research.

---

> > ### Comment · Reviewer_Kinq · 2025-08-06
> >
> > The authors convincingly addressed all four key questions. The authors effectively resolve concerns about module conflicts by detailing standardized interface alignment, presenting concrete dimensional evidence, outlining a structured integration process, and committing to future modular flexibility. These substantive enhancements significantly elevate the paper’s technical depth and practical utility, therefore I will raise the score to 6.

---

> > > ### Author Response · Authors · 2025-08-06
> > >
> > > Thank you very much for your recognition of our work! We truly appreciate the time and effort you dedicated during the review process. Your feedback has been extremely helpful in improving the quality of our paper and our research. Your recognition is highly encouraging and motivating for our future work!

---

### Official Review · Reviewer_PMtS · 2025-06-25

**Rating:** 4
**Confidence:** 4

**Summary:**

This paper presents ModuLM, a framework for Molecular Relational Learning by integrating Large Language Models. The main contributions include offering a flexible and modular approach to model construction, supporting diverse molecular input formats and a wide array of model architectures. ModuLM provides an extensive suite of components, including multiple types of molecular graph encoders, molecular conformation encoders, interaction layers, and mainstream LLM backbones. The framework also supports incremental pretraining strategies to enhance domain-specific capabilities. Comprehensive benchmark experiments demonstrate ModuLM's effectiveness in constructing, evaluating, and comparing LLM-based MRL models.

**Additional Feedback:**

1. Line 58,  "However, integrating these encoding methods into existing LLM frameworks remains a challenge." What is the main challenge? Since there are already some work which use the 2D or 3D encoder first, then do an alighment with LLMs, to make LLMs understand the molecule data.
2.  With other settings the same, a smaller model has a better performance, e.g., 1.8 and 1.10, is it due to the insufficiently fine-tuning the larger model?

**Dataset Code Accessibility:**

Yes

**Dataset Code Comments:**

Dataset and codes are available at https://anonymous.4open.science

**Ethical Considerations:**

No, there are no or only very minor ethics concerns

**Final Justification:**

The rebuttals have clarified the main motivation, and some experimental results, and this work provides an extensive evaluation framework for LLM-based MRL tasks, though it sheds limited new insight.

**Limitations Weaknesses:**

1. This paper focuses primarily on the empirical performance of ModuLM and provides limited theoretical analysis of why certain architectural choices or multimodal integrations are effective.
2. Some of the results are well expected, e.g., models that incorporate 3D molecular conformation information achieve better results. Therefore, more insights are needed for a better MRL.
3. There is a lack of analysis on why ModuLM has better performance as compared to SOTA models such as MolT5, MolCA.

**Strengths Contributions:**

1. ModuLM introduces a unified and flexible framework to support diverse molecular input formats, facilitating the LLM-based MRL.
2. ModuLM can streamline research in MRL by ensuring fair model comparison, and support incremental pretraining and diverse prompt designs, which can enhance the domain-specific capabilities of LLMs in MRL tasks.

---

> ### Author Rebuttal · Authors · 2025-07-29
>
> Thank you for recognizing the flexibility of our ModuLM framework and its ability to facilitate fair model comparison and enhance domain-specific capabilities of LLMs in MRL tasks. Your recognition is greatly appreciated and affirms the value of our work. We will address your questions and concerns one by one below.
>
>
> > **W1.** *The paper  provides limited theoretical analysis of why certain architectural choices or multimodal integrations are effective.*
>
> Thank you for your comment. Our work primarily focuses on providing a fair and extensible evaluation framework to address the current lack of unified performance benchmarks and comparison standards for different encoding methods in LLM-based MRL tasks. As such, our paper places more emphasis on demonstrating the flexibility and scalability of the framework itself, rather than analyzing the impact of individual encoders or input modalities in detail. We believe that the assessment of the effectiveness of specific encoders is the responsibility of the original authors who developed them.
>
> Here, we would like to offer some explanation regarding our architectural design and the effectiveness of our multimodal integration strategy. First, the performance improvement brought by incorporating multiple modalities is intuitive. 2D molecular graphs allow the model to capture more structural features of molecules, while 3D molecular conformations provide detailed spatial information—both of which are beyond the capacity of 1D textual representations alone. For multimodal integration, we adopt a standardized paradigm that is widely used—if not universal—among current multimodal LLMs: concatenating textual and non-textual modalities into a unified input. This strategy has been extensively validated and proven effective by nearly all existing multimodal LLM studies.
>
> Regarding the performance impact brought by different architectural designs, the influence of using different backbones is evident. For different LLMs used as backbones, our input remains consistent—molecular description text and its corresponding multimodal features. The differences in the final inference outputs of these LLMs mainly depend on what they have learned from the input, which is determined by the inherent capabilities of each LLM. Therefore, what we can actually do is focus on how to incorporate more critical molecular structural information into the input. Taking Uni-Mol and PaiNN as examples, when we conduct experiments by keeping all other configurations the same and only replacing the 3D encoder with either Uni-Mol or PaiNN, the superior performance of Uni-Mol can be attributed to its fully-connected Transformer architecture, which allows it to directly model long-range, non-local dependencies between atoms. Additionally, Uni-Mol incorporates 3D positional encoding, enabling the Transformer to effectively capture spatial information. In contrast, although PaiNN aligns more closely with physical principles, it primarily relies on local message passing, which limits its ability to model long-range interactions. As a result, in tasks involving complex molecular structures with non-local chemical environments, its performance is clearly inferior to that of Uni-Mol. This leads to differences in the encoded 3D molecular conformation representations, ultimately causing variations in the downstream reasoning performance when these representations are fed into LLMs. These differences fundamentally stem from the distinct designs of the 3D encoders.
>
>
> > **W2.** *Some of the results are well expected, e.g., models that incorporate 3D molecular conformation information achieve better results. Therefore, more insights are needed for a better MRL.*
>
> Thank you for your valuable comments. The performance improvement brought by incorporating 3D conformations is indeed related to the underlying reaction mechanisms between molecules. The actual reaction process between two molecules essentially involves the binding of their conformations. The 3D conformational information provides richer structural details such as chirality and stereoisomerism, which are crucial for models to better understand molecular interactions and predict reaction mechanisms. In fact, numerous studies have already highlighted the importance of molecular 3D conformations[1],[2]. Therefore, introducing stereochemical information can help models perform deeper reasoning. However, achieving SOTA performance is not the main focus of our paper. Our goal is to develop a flexible framework that supports multimodal fusion and multi-task experiments in MRL. This framework aims to facilitate fair experimental comparison and enable researchers in the field to explore and investigate different combinations and understand why certain combinations are effective, as we have discussed in the limitations and future work sections.
>
>
> [1]Zhu Y, Hwang J, Adams K, et al. Learning Over Molecular Conformer Ensembles. " _The Twelfth International Conference on Learning Representations_.
>
> [2]Li, Sihang, et al. "Towards 3D Molecule-Text Interpretation in Language Models." _The Twelfth International Conference on Learning Representations_.
>
>
>
> > **W3.** *There is a lack of analysis on why ModuLM has better performance as compared to SOTA models such as MolT5, MolCA.*
>
> Thank you for your comments. In fact, this issue has already been addressed in our responses to W1 and W2. First of all, MolT5 and MolCA are not multimodal models—their reasoning relies solely on the textual information of molecules. Therefore, when we incorporate additional multimodal information, enabling LLMs to perceive more structural aspects of molecules, it is evident—especially in the chemical domain—that this leads to significantly improved reasoning capabilities. The integration of 2D molecular graphs and 3D molecular conformations allows LLMs to better understand molecular interactions.
>
> **However, we would like toemphasize that the primary goal of our work is not to achieve SOTA performance or to outperform existing SOTA models. Instead, our objective is to provide an open, extensible, and comparable framework that can facilitate future research on the application of LLMs in MRL.
>
>
> >**F1.** *Line 58, "However, integrating these encoding methods into existing LLM frameworks remains a challenge." What is the main challenge? Since there are already some work which use the 2D or 3D encoder first, then do an alighment with LLMs, to make LLMs understand the molecule data.*
>
> Thank you for your question. While some recent efforts have attempted to integrate both 2D and 3D molecular information into LLMs, there is still a lack of a unified and fair benchmarking platform for comprehensive evaluation. Most existing approaches incorporate only one type of encoder, often tailored to a specific model architecture. This makes it difficult to fairly compare the performance of different encoders or to evaluate how various LLMs perform as backbones under consistent settings. Our work aims to address this gap by providing a flexible and standardized framework for systematic evaluation.
>
>
> >**F2.** *With other settings the same, a smaller model has a better performance, e.g., 1.8 and 1.10, is it due to the insufficiently fine-tuning the larger model?*
>
> Thank you for your question. First, we would like to clarify that in our experiments, we record the model’s performance on both the validation and test sets at each training epoch. For the final evaluation, we report the test performance corresponding to the epoch with the best validation performance. This approach not only allows us to assess model performance effectively but also enables a more detailed analysis of how performance evolves across training epochs under different configurations. In our experimental setup, all models achieve their optimal performance well before reaching the maximum number of training epochs. Therefore, the observed performance differences are not caused by insufficient fine-tuning.
>
> Moreover, this phenomenon has been discussed in existing literature under the concept of inverse scaling[1]. Notably, the _Nature_ article titled "Larger and more instructable language models become less reliable"[2] highlights that, compared to smaller models, larger-scale language models are less likely to refrain from answering questions or tasks that exceed their capabilities. Instead, they tend to produce responses with greater confidence—even when those responses are incorrect. In the context of molecular relational learning, which is a domain less represented in the training data of LLMs compared to general knowledge, larger models may not necessarily perform better. In fact, they might generate more confident yet incorrect predictions. This observation is consistent with our experimental results.
>
> [1] McKenzie I R, Lyzhov A, Pieler M, et al. Inverse scaling: When bigger isn't better[J]. arXiv preprint arXiv:2306.09479, 2023.
>
> [2] Zhou L, Schellaert W, Martínez-Plumed F, et al. Larger and more instructable language models become less reliable[J]. Nature, 2024, 634(8032): 61-68.
>
>
> ---------------
>
> We are deeply grateful for your insightful and helpful comments, which have undoubtedly contributed to improving the quality of our manuscript. If our responses have successfully addressed your concerns and clarified the relevant issues, we sincerely hope that you would consider raising the score. Should you have any further questions or require additional clarification, we would be more than happy to engage in further discussion. Once again, we truly appreciate the time and effort you have devoted to reviewing our manuscript. Your feedback has been invaluable in helping us improve our research.

---

> > ### Comment · Reviewer_PMtS · 2025-08-02
> >
> > Thank the authors for the clarifications, especially on the motivation of this work, and I'll increase my rating to 3.

---

> > > ### Author Response · Authors · 2025-08-02
> > >
> > > Thank you for acknowledging our efforts. We are very pleased to hear that your concerns have been addressed and that you are willing to increase your rating. Should you have additional feedback or concerns, we remain receptive and ready to respond. We are always committed to improving and resolving any remaining issues.
> > >
> > > **It would mean a lot to us if you would kindly consider giving us a positive score!**

---

> > > > ### Author Response · Authors · 2025-08-05
> > > >
> > > > Thank you for your response and for acknowledging that we have addressed your concerns. We would like to check if you might still have any remaining questions or reservations. If so, we would be willing to respond. **We have not noticed any changes to your rating of our manuscript in the system, which appears to be inconsistent with your previous comment.**
> > > >
> > > > **If you would consider further improving your rating of our work, it would be highly encouraging and greatly appreciated as we move forward.**
> > > >
> > > > We sincerely thank you again for the time and effort you have devoted to reviewing our work!

---

> > > > > ### Author Response · Authors · 2025-08-07
> > > > >
> > > > > We have not yet received a further response from you and are unsure whether you have any additional concerns. We would be happy to engage in further discussion if needed. If possible, we kindly hope you could consider giving us a positive score.

---

### Official Review · Reviewer_1U7H · 2025-06-26

**Rating:** 4
**Confidence:** 3

**Summary:**

The submission presents ModuLM, a novel and flexible framework for advancing Molecular Relational Learning (MRL) by integrating large language models (LLMs) with a modular, multimodal architecture. Supporting over 50,000 model configurations, ModuLM allows the use of diverse molecular representations, including 1D SMILES strings, 2D molecular graphs, and 3D conformations. Its key features include a highly extensible modular design, comprehensive benchmarking across tasks such as drug-drug, solute-solvent, and chemical structure interactions, and innovative substructure-based pretraining strategies that enhance molecular understanding. Additionally, ModuLM emphasizes explicit modeling of molecular interactions through interaction-specific prompts and layers, and integrates multiple encoding methods to support generalization across various tasks. Overall, ModuLM stands out as a powerful and versatile tool for modeling complex molecular relationships in drug discovery and beyond.

**Additional Feedback:**

See weaknesss.

**Dataset Code Accessibility:**

Yes

**Ethical Considerations:**

No, there are no or only very minor ethics concerns

**Final Justification:**

My major concerns are addressed, so I keep positive rating.

**Limitations Weaknesses:**

1.Why is the setting of 1.8 in Table 4 and Table 5 more effective than the setting of 1.10? Please analyze. According to the settings in Table 3, the backbone used in 1.10 is larger, why is the result actually worse?

2.In Table 3, the input features are provided in only one or two forms. Does the framework support simultaneously using all three types of input features: m_s, m_g, and m_c?

**Strengths Contributions:**

1.Modular and Extensible Design: ModuLM features a highly modular architecture that allows users to flexibly assemble and extend models tailored to specific molecular relational learning (MRL) tasks. It supports various molecular representations—1D SMILES, 2D molecular graphs, and 3D conformations—along with customizable prompt mechanisms and multimodal integration strategies, making it adaptable across diverse input types.

2.Clear Writing and Logical Organization: The paper is well-written and logically structured, presenting the problem formulation, methodology, experiments, and results in a coherent manner.

3.Well-Justified Distinction from Prior Work: The authors provide thorough comparisons with existing models and tools, clearly articulating how ModuLM advances the field. Its unique modularity, multimodal adaptability, and support for customized interaction modeling distinguish it from prior approaches.

---

> ### Author Rebuttal · Authors · 2025-07-29
>
> Thank you very much for your positive and encouraging feedback. We sincerely appreciate your recognition of ModuLM’s modular and extensible design, the clarity and organization of our writing, and the distinctions we drew between our framework and prior work. Your acknowledgment of our efforts to support multimodal molecular representations and comprehensive interaction modeling is highly motivating！Below, I will address some of the questions and concerns you raised.
>
>
> > **W1.** *Why does using a larger-scale LLM as the backbone result in worse performance?*
>
> Thank you for your question. Indeed, this is a counterintuitive phenomenon, but it has been observed and discussed in recent research[1],[2] . Notably, the Nature article titled "Larger and more instructable language models become less reliable" [2]points out that compared to smaller models, larger-scale language models are less likely to avoid questions or tasks beyond their capabilities. As a result, they tend to respond with greater confidence—even when their answers are incorrect. In the context of molecular relational learning, which is a domain less represented in LLM training data compared to more common knowledge areas, larger models may not necessarily perform better. In fact, they may generate more confident yet incorrect responses. This aligns with our experimental results, where smaller models sometimes outperform their larger counterparts on MRL tasks.
>
> [1] McKenzie I R, Lyzhov A, Pieler M, et al. Inverse scaling: When bigger isn't better[J]. arXiv preprint arXiv:2306.09479, 2023.
>
> [2] Zhou L, Schellaert W, Martínez-Plumed F, et al. Larger and more instructable language models become less reliable[J]. Nature, 2024, 634(8032): 61-68.
>
> > **W2.** *Does the framework support simultaneously using all three types of input features: m_s, m_g, and m_c?*
>
> Thank you for your feedback. In fact, our framework supports flexible combinations of the three modalities: text, molecular graphs, and molecular conformations. However, since our backbone is based on language models, we typically do not omit the text modality. We have provided additional experimental results under configurations that utilize all three modalities. Here, we use **Accuracy** and **RMSE** for DDI and SSI as the evaluation metrics, respectively. It is worth noting that when using both 2D and 3D modalities simultaneously, adopting the Qformer alignment approach requires setting up a separate Qformer for each of the 2D and 3D modalities. This significantly slows down the training speed of the model without providing noticeable performance improvements. Therefore, we do not recommend using both 2D and 3D modalities for representation at the same time.
>
> | Configuration\Datasets          | ZhangDDI | ChChMiner | FreeSolv | CombiSolv |
> | ------------------------------- | -------- | --------- | -------- | --------- |
> | LLama1b+GIN+Unimol+MLP          | 87.7%    | 94.2%     | 0.862    | 0.762     |
> | deepseek1.5b+GIN+Unimol+MLP     | 91.0%    | 96.5%     | 0.689    | 0.380     |
> | deepseek1.5b+MPNN+SchNet+MLP    | 90.2%    | 95.6%     | 0.722    | 0.401     |
> | deepseek1.5b+GIN+Unimol+Qformer | 91.3%    | 96.1%     | 0.672    | 0.374     |
>
> Although we are unable to conduct more extensive experiments, the existing data indicate that incorporating both 2D and 3D information does not provide a significant performance improvement compared to using only 3D information.
>
> ---------------
>
> Thank you once again for your recognition of our work. We are truly grateful for your insightful and helpful comments, which have undoubtedly contributed to improving the quality of our manuscript. If our response has successfully addressed your concerns and clarified any ambiguities, we respectfully hope that you would consider raising the score. Should you have any further questions or require additional clarification, we would be more than happy to engage in further discussion. Once again, we sincerely appreciate the time and effort you have devoted to reviewing our manuscript. Your feedback has been invaluable in enhancing our research.

---

> > ### Author Response · Authors · 2025-08-05
> >
> > Thank you very much for your positive recognition of our work. We have carefully addressed your comments and provided detailed responses. We hope our revisions have adequately resolved your concerns. If there are any remaining questions or suggestions, we would be more than happy to respond.**If your concerns have been fully addressed, we sincerely look forward to your feedback and would greatly appreciate it if you could  improve your rating of our manuscript.**
> >
> > Thank you once again for your time and valuable input during the review process!

---

> > ### Comment · Reviewer_1U7H · 2025-08-08
> >
> > Thanks for the rebuttal, and I will keep my positive rating.

---

> > > ### Author Response · Authors · 2025-08-09
> > >
> > > We sincerely appreciate your recognition of our article and the effort you devoted during the review process! Your comments have been extremely helpful to our work!

---

### Official Review · Reviewer_qmBc · 2025-06-30

**Rating:** 4
**Confidence:** 4

**Summary:**

This paper proposes ModuLM, a modular framework for constructing and evaluating large language model (LLM)-based molecular relational learning (MRL) models. The framework addresses key challenges in the field by providing flexible support for multiple molecular input formats (1D SMILES, 2D graphs, 3D conformations), diverse encoder architectures, and various LLM backbones. ModuLM enables dynamic construction of over 50,000 distinct model configurations through its modular design, supporting 8 types of 2D molecular graph encoders, 11 types of 3D molecular conformation encoders, 7 interaction layers, and 7 mainstream LLM backbones. The authors demonstrate the framework's effectiveness across multiple MRL tasks including drug-drug interactions (DDI), solute-solvent interactions (SSI), and chromophore-solvent interactions (CSI).

**Additional Feedback:**

- Include comparisons with Geoformer, GotenNet, and other recent SOTA molecular representation learning methods to provide a complete picture of performance.
- How does the framework handle molecular size limitations across different encoders?
- How does performance scale with molecular complexity and dataset size?

**Dataset Code Accessibility:**

Yes

**Dataset Code Comments:**

Data loaders and the code is available and well documented.

**Ethical Considerations:**

No, there are no or only very minor ethics concerns

**Final Justification:**

Authors made important clarifications.

**Limitations Weaknesses:**

- The three pretraining strategies (molecular interaction-based, substructure-based, structure similarity-guided grouping) are relatively straightforward extensions of existing approaches.
- The paper lacks comparison with several important recent state-of-the-art methods in molecular representation learning, particularly Geoformer and GotenNet, which have shown strong performance on molecular tasks. This significantly undermines the comprehensiveness of the evaluation.
- Limited analysis of computational efficiency and scalability, which are crucial for practical adoption.
- Limited discussion of how the framework handles potential overfitting when exploring such a large configuration space.

**Strengths Contributions:**

- The modular architecture is well-conceived, addressing a genuine need for standardized benchmarking in LLM-based MRL. The ability to generate 50,000+ model configurations demonstrates impressive flexibility.
- Comprehensive support for multimodal molecular representations (1D/2D/3D) fills an important gap, particularly the inclusion of 3D conformational data which is often overlooked in LLM-based approaches.
- The framework includes important components often missing from existing systems: explicit interaction modeling layers, multiple pretraining strategies, and support for chain-of-thought reasoning.
- Clear demonstration that multimodal information (especially 3D conformations) consistently improves performance across tasks and model architectures.

---

> ### Author Rebuttal · Authors · 2025-07-29
>
> Thank you very much for your valuable comments. We are greatly encouraged by your recognition of the ModuLM framework design as well as the flexibility of its modalities and modules. Below, we provide point-by-point responses to your suggestions and concerns.
>
>
> > **W1.** *The three pretraining strategies are elatively straightforward extensions of existing approaches.*
>
> Thank you very much for your valuable feedback. Our primary goal is to develop a fair, unified, and user-friendly evaluation framework for LLM-based molecular relational learning (MRL), with a key focus on systematically integrating various pretraining strategies. Notably, most existing LLM-based MRL studies tend to overlook the incremental pretraining step, which we consider essential for enhancing downstream performance. Our work highlights this often-neglected phase. Beyond incorporating mainstream approaches, we also propose a novel grouping-based incremental pretraining strategy guided by structural similarity—a direction, to the best of our knowledge, not yet explored in prior research. This contribution not only extends existing methods but also offers a new perspective for incremental pretraining in LLM-based molecular learning.
>
> > **W2&F1.** *Lacking comparisons with recent advanced molecular representation methods, such as Geoformer and GotenNet.*
>
> Thank you very much for your thoughtful feedback. First, we would like to clarify that our work focuses on **molecular relation learning (MRL)** rather than molecular representation learning. Second，I believe what you referred to here might be **GeoMFormer**[1], rather than **Geoformer**[2]. The primary goal of our study is to provide a convenient and extensible benchmark for evaluating LLMs in the context of MRL tasks. We compared our approach against MolTC[3], which was the leading model in molecular relation learning benchmarks at the time of submission. In response to your suggestion, we have further incorporated the molecular representation learning models GeoMFormer[1] and GotenNet[4] into our 3D encoding module and conducted comparative experiments under a subset of settings. Due to time constraints, the current evaluation covers only partial configurations, but we believe it already offers valuable insights into their relative performance within our framework.
>
> Here, we evaluated two datasets from the DDI task and two datasets from the SSI task, using **Accuracy** and **RMSE** as the evaluation metrics, respectively. Notebly, to ensure comparability with the experimental results in the main text, we only replaced the encoding module while keeping all other configurations unchanged.
>
> | Configuration\Datasets  | ZhangDDI | ChChMiner | FreeSolv | CombiSolv |
> | ----------------------- | -------- | --------- | -------- | --------- |
> | deepseek1.5b+GeoMFormer | 0.911    | 0.962     | 0.680    | 0.374     |
> | deepseek1.5b+GotenNet   | 0.918    | 0.965     | 0.649    | 0.352     |
>
> [1] Chen T, Luo S, He D, et al. GeoMFormer: A general architecture for geometric molecular representation learning[J]. arXiv preprint arXiv:2406.16853, 2024.
>
> [2]Yu J, Huang B, Zhang Y, et al. Geoformer: Learning point cloud completion with tri-plane integrated transformer[C]//Proceedings of the 32nd ACM International Conference on Multimedia. 2024: 8952-8961.
>
> [3] Fang J, Zhang S, Wu C, et al. Moltc: Towards molecular relational modeling in language models[J]. arXiv preprint arXiv:2402.03781, 2024.
>
> [4] Aykent S, Xia T. Gotennet: Rethinking efficient 3d equivariant graph neural networks[C]The Thirteenth International Conference on Learning Representations. 2025.
>
>
> > **W3.** *Limited analysis of computational efficiency and scalability.*
>
> Thank you for your valuable feedback. Regarding computational efficiency, we have provided detailed runtime statistics in the table and will include them in the revised version of the paper. As the results show, the main sources of computational cost arise from the integration of diverse molecular modalities and the inherent complexity of LLMs. In addition, the alignment mechanism used in Qformer also contributes to increased computational overhead.To accommodate future users who may incorporate even larger models, we have included support for distributed training in our codebase. This can be easily enabled by adding just one line in the configuration file, making the system readily scalable to more demanding setups.
>
> Here, we provide the number of samples processed per second on a single GPU as a reference for computational throughput, based on an 8×A800 GPU setup.
>
> | deepseek1.5b                    | deepseek1.5b+GIN+MLP          | Galactica1.3b+GIN+MLP           |
> | ------------------------------- | ----------------------------- | ------------------------------- |
> | 4.22it/s                        | 3.38it/s                      | 3.16it/s                        |
> | **deepseek1.5b+GIN+Qformer**    | **deepseek1.5b+Unimol+MLP**   | **deepseek1.5b+Unimol+Qformer** |
> | 2.98it/s                        | 2.74it/s                      | 2.28it/s                        |
> | **deepseek7b+Unimol+MLP**       | **Galactica1.3b+Unimol+MLP**  | **LLama1b+Unimol+MLP**          |
> | 1.98it/s                        | 2.45it/s                      | 2.40it/s                        |
> | **deepseek1.5b+GeoMFormer+MLP** | **deepseek1.5b+GotenNet+MLP** | **deepseek1.5b+MPNN+MLP**       |
> | 2.68it/s                        | 2.86it/s                      | 3.20it/s                        |
>
> As for scalability, Section 4.3 of the paper demonstrates ModuLM’s extensibility by integrating custom modules and evaluating their performance within the framework. ModuLM is designed with a modular, Lego-like philosophy—users can flexibly assemble their desired models using the components we provide, or incorporate new modules with minimal effort. Therefore, ModuLM offers strong scalability and adaptability, both architecturally and functionally.
>
> **W4.** *Limited discussion of how the framework handles potential overfitting when exploring such a large configuration space.*
>
> Thank you for your feedback. Since our work serves as a benchmark evaluation framework, we do not apply overfitting prevention techniques such as early stopping, in order to better analyze how model performance evolves over training epochs. All model configurations follow their original implementations as described in the respective papers as much as possible. During the experiments, we record the model’s performance on both the validation and test sets at each epoch. When reporting final results, we select the test performance corresponding to the best validation performance, which can be regarded as a manual early stopping strategy to mitigate overfitting. This approach allows us to analyze model performance from multiple perspectives.
>
> **F2&F3.** *How does the framework handle molecular size limitations across different encoders and how does performance scale with molecular complexity and dataset size?*
>
> Thank you for your question. In fact, our work aims to provide a fair and convenient evaluation tool for applying LLMs to MRL tasks. We focus on offering a framework that integrates various types of encoders. To ensure fair evaluation across molecules of different sizes, we do not apply any special dataset processing tailored to specific encoders. In practice, large molecules account for only a small proportion of the datasets we use. However, we appreciate your suggestion, these analysis can be easily enabled by making minor modifications to our data preprocessing step. We provide below some experimental results showing model performance under different levels of molecular complexity and dataset size for selected configurations.
>
> We integrated ZhangDDI, ChChDDI, DeepDDI, and Twosides into a large dataset and grouped the data based on molecular mass, with 20,000 samples in each group. AM represents the average molecular mass. We use Accuracy as the evaluation metric.
>
> | Configuration\AM            | 334    | 566    | 628    | 730    | 1846   |
> | --------------------------- | ------ | ------ | ------ | ------ | ------ |
> | deepseek1.5b+GIN+MLP        | 88.52% | 87.30% | 86.04% | 84.56% | 73.42% |
> | deepseek1.5b+Unimol+MLP     | 89.12% | 88.42% | 87.34% | 84.64% | 75.32% |
> | LLama1b+Unimol+MLP          | 85.68% | 83.55% | 82.98% | 80.40% | 70.88% |
> | deepseek1.5b+Unimol+Qformer | 89.31% | 88.50% | 87.48% | 84.42% | 74.98% |
> | deepseek1.5b+GIN+Qformer    | 88.46% | 87.60% | 85.56% | 84.13% | 73.12% |
>
> To facilitate a better comparison with the experiments in the main text, we select the ZhangDDI dataset and divide it into different scales for experimentation. We continue to use accuracy as the evaluation metric.  Due to the limitations of multimodal models, we are unable to perform zero-shot testing on them, which would lead to a complete collapse in model performance.
>
> | Configuration\Size          | 1000   | 5000   | 10000  | 20000  | 50000  |
> | --------------------------- | ------ | ------ | ------ | ------ | ------ |
> | deepseek1.5b+GIN+MLP        | 67.32% | 75.65% | 81.82% | 86.72% | 88.31% |
> | deepseek1.5b+Unimol+MLP     | 70.03% | 77.38% | 82.63% | 87.33% | 90.13% |
> | LLama1b+Unimol+MLP          | 64.57% | 73.20% | 78.95% | 83.28% | 86.64% |
> | deepseek1.5b+Unimol+Qformer | 68.80% | 76.32% | 82.34% | 87.28% | 90.20% |
> | deepseek1.5b+GIN+Qformer    | 66.12% | 74.88% | 81.46% | 86.80% | 88.67% |
>
> ---------------
>
> Thank you for your valuable feedback. We conducted additional experiments to address your concerns, but due to time constraints, we couldn’t test all configurations mentioned in the paper. We hope our responses clarify your questions and would appreciate your consideration for a higher score. If you have further questions or need more details, we’re happy to discuss. Thank you again for your time and helpful comments, which have greatly improved our work.

---

> > ### Author Response · Authors · 2025-08-05
> >
> > We have carefully addressed each of your comments and provided detailed responses. We hope our revisions have adequately resolved your concerns. If there are any remaining questions or suggestions, we would be willing to respond.**If you find that your concerns have been fully addressed, we would greatly appreciate your feedback and hope that you could consider improving your rating of our work.**
> >
> > Thank you once again for your time and valuable input during the review process!

---

> > ### Comment · Reviewer_qmBc · 2025-08-06
> >
> > Thank you to the authors for their thorough and responsive rebuttal. I appreciate the significant effort demonstrated by the additional experiments conducted during the rebuttal period. The new results comparing ModuLM with GeoMFormer and GotenNet, along with the detailed analyses on computational efficiency and performance scaling with respect to molecular complexity and dataset size, have successfully addressed all of my primary concerns. As a result, I have raised my score to 4. I trust that the authors will incorporate these valuable new experiments, tables, and discussions into the final version of the paper, as they significantly strengthen the work.

---

> > > ### Author Response · Authors · 2025-08-07
> > >
> > > Thank you for acknowledging our efforts. We are very pleased to hear that your concerns have been addressed and that you are willing to increase your rating. We will include the additional experiments based on your suggestions in the final version of the paper. **Thank you for the time and effort you dedicated to reviewing our work, it has been tremendously helpful to us!**

---

### Note · Authors · 2025-08-12

Dear AC, SAC, and PC,

Thank you for your valuable efforts in reviewing our work; your feedback has greatly helped us improve it. Below is a summary of the discussion and key strengths of our work as recognized by the reviewers.

**Discussion Summary**
- **Concerns on Efficiency, Dataset Scale, and Encoder Impact.** We responded with more comprehensive experiments and detailed analyses, which received **positive feedback**(**Reviewer qmBc**).
- **Experimental Setup and Result Analysis.** We provided additional clarifications and further analyzed the results, which received **positive feedback** (**Reviewers 1U7H, Kinq**).
- **Theoretical Analysis of Performance Differences.** We provided additional discussion and explored possible causes, which received the reviewer’s recognition and led to an increased rating(**Reviewer PMtS**).
- We comprehensively addressed reviewer's concerns and received recognition from **all the reviewers**. All reviewers did not raise any further questions during the discussion.

**Highlights of the work**
- **Highly Modular and Extensible Design.** The framework features a modular architecture that enables flexible assembly and extension of models for diverse MRL tasks (**Reviewers qmBc, 1U7H, PMtS, Kinq**).
- **Multi-Modality Support and Flexibility**. Supports 1D/2D/3D multimodal molecular inputs, notably including 3D conformational data (often overlooked in prior LLM approaches), with customizable prompting mechanisms and multimodal integration strategies (**Reviewers qmBc, 1U7H, Kinq**).
- **Systematic and Standardized Evaluation Capability.** Capable of generating 50,000+ distinct model configurations, filling a gap in systematic evaluation tools for the field and enabling fair model comparisons (**Reviewers qmBc, 1U7H, PMtS**).
- **Thorough Experiments and Clear Presentation.** Conducts comprehensive experiments across multiple MRL tasks; the paper is clearly structured (**Reviewers qmBc, 1U7H, Kinq**).

We also proposed a SOTA model in our work, but this was intended to demonstrate that **ModuLM can lay a solid foundation for deeper and more comprehensive exploration of LLMs in MRL, rather than merely to improve performance metrics**,as we have discussed in our response to **Reviewer PMtS**.

We sincerely appreciate your patience and effort in reviewing our work. **We truly value the opportunity to present our research at NeurIPS and earnestly hope for the AC’s understanding and support!**

Best regards,
Authors

---

### Decision · Program_Chairs · 2025-09-18

**Decision:**

Accept (poster)

**Comment:**

In this submission, the authors developed a highly compatible benchmark for LLM-based molecular relational learning. The benchmark provides a highly flexible I/O interface to support various data formats. In addition, various LLMs can be applied as backbone models. All four reviewers evaluated this work positively, and the authors made efforts to resolve the concerns about the potential limitations of the benchmark (e.g., what are the suggested configurations given so many choices?). In summary, I think this benchmark contributes to the community of MRL and tend to accept this work.